# Inhibiting the integrated stress response pathway prevents aberrant chondrocyte differentiation thereby alleviating chondrodysplasia

Cheng Wang[1†], Zhijia Tan[1†], Ben Niu[1], Kwok Yeung Tsang[1], Andrew Tai[1], Wilson C W Chan[1], Rebecca L K Lo[1], Keith K H Leung[1], Nelson W F Dung[1], Nobuyuki Itoh[2], Michael Q Zhang[3,4], Danny Chan[1], Kathryn Song Eng Cheah[1*]

[1]School of Biomedical Sciences, University of Hong Kong, Hong Kong, China; [2]Graduate School of Pharmaceutical Sciences, University of Kyoto, Kyoto, Japan; [3]Department of Biological Sciences, Center for Systems Biology, The University of Texas at Dallas, Richardson, United States; [4]MOE Key Laboratory of Bioinformatics, Center for Synthetic and Systems Biology, Tsinghua University, Beijing, China

**Abstract** The integrated stress response (ISR) is activated by diverse forms of cellular stress, including endoplasmic reticulum (ER) stress, and is associated with diseases. However, the molecular mechanism(s) whereby the ISR impacts on differentiation is incompletely understood. Here, we exploited a mouse model of Metaphyseal Chondrodysplasia type Schmid (MCDS) to provide insight into the impact of the ISR on cell fate. We show the protein kinase RNA-like ER kinase (PERK) pathway that mediates preferential synthesis of ATF4 and CHOP, dominates in causing dysplasia by reverting chondrocyte differentiation via ATF4-directed transactivation of *Sox9*. Chondrocyte survival is enabled, cell autonomously, by CHOP and dual CHOP-ATF4 transactivation of *Fgf21*. Treatment of mutant mice with a chemical inhibitor of PERK signaling prevents the differentiation defects and ameliorates chondrodysplasia. By preventing aberrant differentiation, titrated inhibition of the ISR emerges as a rationale therapeutic strategy for stress-induced skeletal disorders.
DOI: https://doi.org/10.7554/eLife.37673.001

**\*For correspondence:**
kathycheah@hku.hk

[†]These authors contributed equally to this work

**Competing interests:** The authors declare that no competing interests exist.

## Introduction

The Integrated Stress Response (ISR) is a eukaryotic cellular stress response, aiming to restore cellular homeostasis upon different types of extrinsic or intrinsic stresses. The ISR can be stimulated by a range of physiological or pathological changes (*Brostrom et al., 1996*; *Dever et al., 1992*; *Harding et al., 2003*; *Ron, 2002*; *Wek et al., 2006*), including hypoxia, amino acid deprivation, glucose/nutrition deprivation, viral infection (*Dever et al., 1992*; *Harding et al., 2003*; *Wek et al., 2006*; *García et al., 2007*; *Rzymski et al., 2010*; *Ye et al., 2010*) and intrinsic endoplasmic reticulum (ER) stress (*Harding et al., 1999*), which is caused by the accumulation of unfolded or misfolded proteins within ER. Furthermore, in the context of cancer biology, oncogene activation can also trigger the ISR (*Denoyelle et al., 2006*; *Hart et al., 2012*). Although the ISR is primarily a pro-survival homeostatic program, aiming to optimize the adaptive cellular response to stress, exposure to severe stress, either in intensity or duration, will overwhelm the capacity of this adaptive response and drive signaling toward cell death.

The key early controlling step in the ISR is the phosphorylation of eukaryotic translation initiation factor 2 alpha (eIF2α) by one of four members of eIF2α kinase family: protein kinase R (PKR), PKR-

like endoplasmic reticulum kinase (PERK), general control nonderepressible 2 (GCN2), and Heme-regulated eIF2α kinase (HRI) (*Dever et al., 1992*; *García et al., 2007*; *Harding et al., 1999*; *Han et al., 2001*). These kinases phosphorylate serine 51 in eIF2α, promoting the formation of a p-eIF2α and eIF2B complex, consequently inhibiting the guanine nucleotide exchange activity of eIF2B (*Sudhakar et al., 2000*). Inactivation of the eIF2 complex leads to a shutdown of global protein synthesis but also the induction of preferential translation of transcripts notably the mRNAs of transcription factors, Activating Transcription Factor 4 (ATF4) and C/EBP homologous protein (CHOP, encoded by *Ddit3*), and other factors with both pro-survival and pro-death functions which aid cell adaption and recovery (*Figure 1—figure supplement 1A*) (*Harding et al., 2000*). eIF2α phosphorylation is transient and can be reversed by PPP1R15A (GADD34) or CReP1, the regulatory subunit of eIF2α phosphatases, acting in a negative feedback loop that allows protein synthesis to restart (*Harding et al., 2009*; *Novoa et al., 2001*). When the stress is intense or prolonged, cells fail to adapt, and apoptosis is triggered. It is likely that the duration and level of eIF2α phosphorylation, as well as ATF4 regulation, determine the balance between cell survival and cell death.

There are more than 400 human genetic skeletal disorders caused by disrupted cartilage and bone development and growth, commonly resulting in dwarfism and skeletal deformities (*Geister and Camper, 2015*). Many of these disorders are caused by mutations in genes for extracellular matrix (ECM) proteins (*Geister and Camper, 2015*; *Bonafe et al., 2015*; *Briggs and Chapman, 2002*; *Warman et al., 1993*), and some mutations cause inappropriate folding, processing or export, leading to retention in the ER, affecting the secretory and stress response pathways (*Wilson et al., 2005*; *Tsang et al., 2007*; *Tsang et al., 2010*; *Posey et al., 2012*; *PirogPiróg-Garcia et al., 2007*; *Hartley et al., 2013*; *Cameron et al., 2011*; *Boot-Handford and Briggs, 2010*; *Arnold and Fertala, 2013*). Accumulation of misfolded proteins in the ER can overwhelm the protein quality control mechanism, causing proteotoxicity, cell cycle arrest and cell death. Cells activate the unfolded protein response (UPR), which mediates cell survival by slowing protein translation, promoting proteostasis via the proteasome and activating transcription factors that upregulate the production of protein chaperones (reviewed in [*Hotamisligil and Davis, 2016*; *Horiuchi et al., 2016*]). The UPR employs three arms of sensors in the ER to mediate cell adaptation and survival under ER stress: the key component of the ISR, PERK, inositol-regulated enzyme 1α (IRE1α), and the activating transcription factor 6 (ATF6) family (*Hotamisligil and Davis, 2016*; *Horiuchi et al., 2016*; *Walter and Ron, 2011*). Upon ER stress, the PERK-p-eIF2α signaling modulates the cell adaptation via translational control. ATF6 family factors move from the ER to the Golgi, are processed by S1 and S2 proteases, and translocate to the nucleus to activate ER quality control genes such as *Hspa5* (encodes BiP) and *Xbp1* (X-box binding protein 1). IRE1α has kinase and endoribonuclease (RNase) activities. It catalyzes the splicing of *Xbp1* mRNA, generating the UPR transcription factor XBP1[S] that upregulates genes encoding chaperones and proteins involved in ER-associated protein degradation (ERAD).

Human metaphyseal chondrodysplasia type Schmid (MCDS; MIM156500) is an autosomal dominant disorder caused by heterozygous mutations in the NC1 domain of type X collagen, encoded by *COL10A1* in hypertrophic chondrocytes (HC) (*Warman et al., 1993*; *Wilson et al., 2005*; *Mäkitie et al., 2005*). In an MCDS transgenic mouse model (13del), carrying a 13 bp deletion in *Col10a1* equivalent to the human mutation, misfolded mutant collagen X induces ER stress suggesting the primary role of ER stress in MCDS pathogenesis (*Tsang et al., 2007*). Although the chondrocytes survive, their differentiation is reversed by an unknown mechanism to a more juvenile state characterized by the re-expression of prehypertrophic chondrocyte markers (*Ppr*, *Sox9* and *Col2a1*), disrupting endochondral ossification, and skeletal dysplasia ensues. The causative role of ER stress in MCDS is further supported by studies showing that expression of an exogenous misfolded protein in HCs can induce an MCDS-like phenotype (*Rajpar et al., 2009*).

It is noteworthy that the skeletal defects caused by mutations that induce stress or inactivate critical transducers of the stress response in humans (*Julier and Nicolino, 2010*) and mouse models (*Tsang et al., 2007*; *Cameron et al., 2011*; *Horiuchi et al., 2016*; *Rajpar et al., 2009*; *Julier and Nicolino, 2010*; *Nundlall et al., 2010*; *Cameron et al., 2015a2015*; *Cameron et al., 2015b*) strongly implicate components of ER stress-induced UPR pathways involved in chondrocyte and osteoblast differentiation (*Tsang et al., 2010*; *Horiuchi et al., 2016*). Expression of misfolded cartilage oligomeric matrix protein (COMP) in proliferating and hypertrophic chondrocytes results in chondrocyte disorganization and causes Pseudoachondroplasia (PSACH), suggesting triggering the

UPR may underlie many chondrodysplasias where mutations cause accumulation of misfolded proteins (*Posey et al., 2012*; *Piróg et al., 2014*). Another example that links ER stress signaling to chondrocyte differentiation is illustrated by studies on BBF2H7, an ER stress transducer. *Bbf2h7* null mutants show severe chondrocyte abnormality due to proliferation and differentiation defects, indicating its essential role for chondrogenesis (*Saito et al., 2009*; *Saito et al., 2014*). Furthermore, BBF2H7 has been shown to suppress chondrocyte hypertrophy by direct regulation of IHH signaling and the IHH-PTHrP pathway (*Saito et al., 2014*).

Pharmacological stimulation of intracellular proteolysis of mutant collagen X in another MCDS mouse model reduces the level of ER stress, partially ameliorating the dwarfism phenotype (*Mullan et al., 2017*). However, the rescue of the chondrocyte differentiation defect, expansion of the hypertrophic zone and bone growth was incomplete, suggesting stimulating degradation of misfolded protein alone is not sufficient to address the impact of the ER stress on aberrant cell differentiation.

The relative contribution of the arms of the UPR and its constituent components to the pathology and a molecular understanding of the consequences of activation of ER stress on cell fate and differentiation *in vivo* is lacking. Here by studying MCDS in a mouse model, we sought to clarify the mechanism(s) by which the UPR/ISR causes aberrant differentiation defects which may provide the basis for the development of novel therapeutic target(s) for treating this disorder, or other ER stress-related skeletal disorders. We reveal by genetic, biochemical and *in vivo* functional approaches in mouse models, ISR-mediated preferential translation of ATF4 directly activates inappropriate expression of the key transcription factor SOX9. This ectopic SOX9 expression in HCs reverts chondrocyte differentiation, thereby causing MCDS. By targeting the ISR early, at the level of p-eIF2α induction of ATF4, we ameliorate the pathology, thereby providing a rationale pharmacological strategy for treating MCDS and other skeletal disorders caused by activation of the ISR.

## Results

### The ER stress-induced UPR disrupts global transcriptome patterns in the chondrodysplastic growth plate

The mammalian growth plate comprises four major sub-populations of chondrocytes: resting, proliferating (PC), prehypertrophic (pHC) and hypertrophic chondrocytes (HC). These chondrocytes have distinct morphologies and gene expression profiles governed by a precisely tuned gene regulatory network (*Hojo et al., 2016*). To investigate the effect of ER stress on the transcriptome of HCs, the proximal tibial growth plates from 10-day-old WT and 13del mice were fractionated into sub-populations representing proliferating (PZ), prehypertrophic (pHZ) and hypertrophic chondrocytes (HZ) (*Figure 1A*; *Figure 1—figure supplement 1B*). The wild-type HZ was fractionated into upper and lower zones (UHZ and LHZ) to capture early onset and late phases of hypertrophy. The *13del* HZ was fractionated into three zones: upper (UHZ) corresponding to the early phase of UPR activation, middle (MHZ) where HC adaptation would be initiated, and lower (LHZ) where HC should fully adapt.

We used *k*-means clustering (see Materials and methods) to categorize the gene expression patterns across different zones in wild-type and 13del growth plates into four clusters (*Supplementary file 1*). Genes (453) in Cluster I increased expression from PHZ to lower HZ specifically in 13del HC (*Figure 1B and C*). Ontological analyses show these differentially expressed genes are mainly involved in protein processing in the ER and the UPR (*Figure 1D and E*, *Supplementary file 2* and *3*). Genes in Clusters II (659) and III (314) showed highest expression in wild-type PZ and pHZ followed by progressive downregulation from pHZ to LHZ but were upregulated in 13del LHZ, reflecting UPR-induced changes in HC differentiation (*Figure 1B and C*). These genes included *Sox9*, *Ppr*, and *Ihh*, consistent with our previous report of re-expression of pre-hypertrophic markers (*Tsang et al., 2007*). Cluster IV genes (680) showed increasing expression from pHZ to LHZ in wild-type and can be defined as 'hypertrophy characteristic' genes. Consistent with a change in the HC de-differentiation state in 13del, these genes were down-regulated in 13del LHZ. The concomitant down-regulation of Cluster I stress response genes in 13del LHZ is consistent with the alleviation of the stress in the reprogrammed cells and an adapted state.

We further compared our microarray dataset with the published data (*Cameron et al., 2011*) from another 2 MCDS mouse models, expressing a Col10a1 p.N617K mutation or an ER stress-

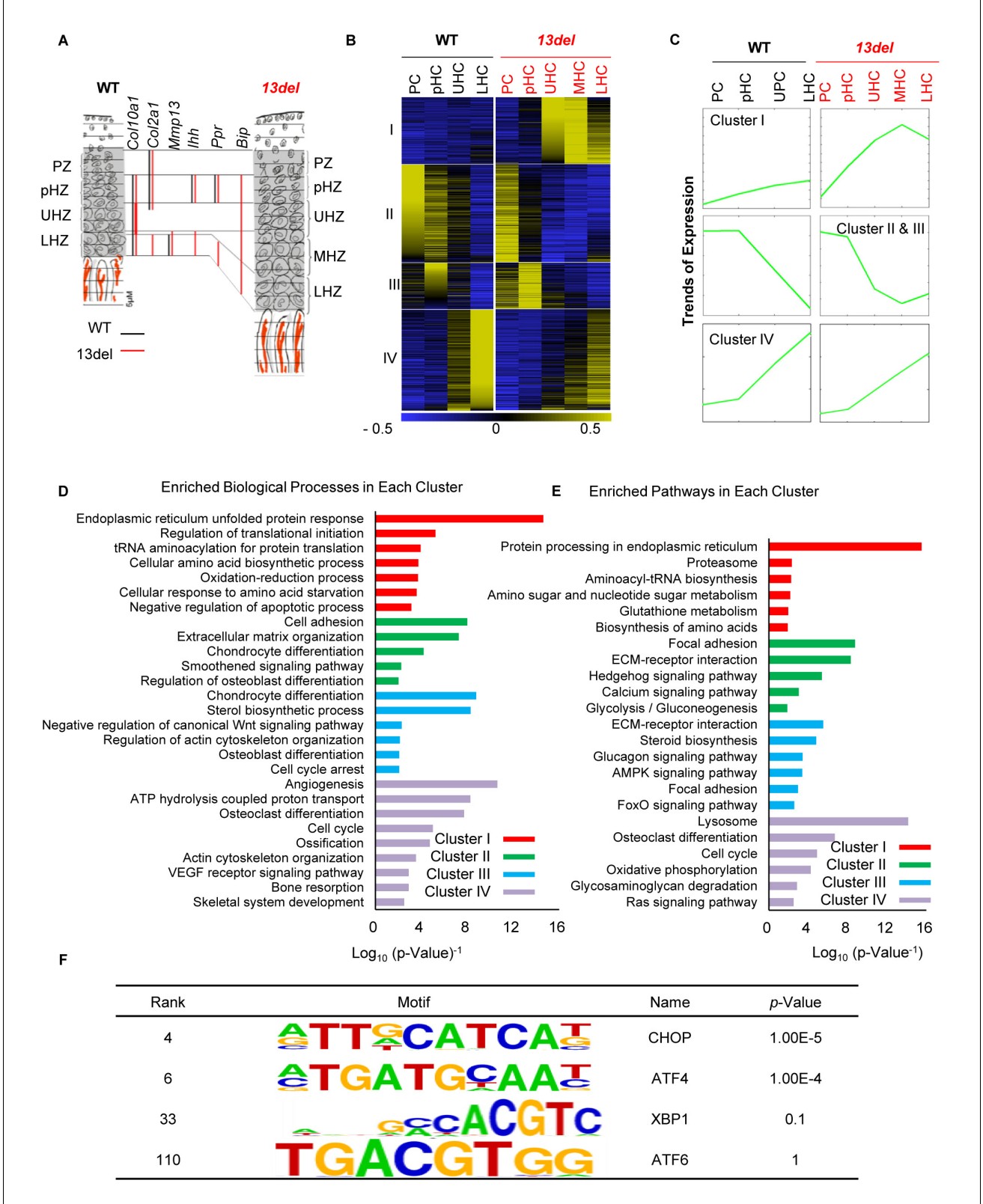

**Figure 1.** PERK signaling pathway, activated by ER stress, plays an etiological role in MCDS *13del* mice. (**A**) Schematic diagram of the rationale for fractionating the WT and 13del p10 growth plates into different chondrocyte populations. (**B**) Clustering analysis of differentially expressed genes in chondrocyte subpopulations in p10 WT and 13del proximal tibial growth plates. Expression levels were normalized from −0.5 (blue) to 0.5 (yellow). Four major clusters were identified. (**C**) The average expression levels (Log$_2$ scale) of the genes in different clusters revealed significant expression pattern

*Figure 1 continued on next page*

*Figure 1 continued*

changes in 13del mice. (D–E) Genes in different clusters were functionally categorized using DAVID web tools. The enriched biological processes (D) and enriched pathways (E) were sequentially shown for Cluster I, II, III and IV. The values on the X-axis represented the $\log_{10}$ ($p$-value$^{-1}$). Each category with p-value <0.05 was considered as significantly enriched. (F) Enriched motifs on Cluster I genes were identified, using sequences of the promoter region (±2 kb from the TSS) for these genes. Motifs matched to the TFs in the UPR were shown.

DOI: https://doi.org/10.7554/eLife.37673.002

The following figure supplements are available for figure 1:

**Figure supplement 1.** Transcriptome analyses of 13del chondrocytes.

DOI: https://doi.org/10.7554/eLife.37673.003

**Figure supplement 2.** Activation of PERK pathway in *13del* Mice.

DOI: https://doi.org/10.7554/eLife.37673.004

**Figure supplement 3.** Ectopic expression of XBP1$^s$ in HCs does not cause any abnormalities in growth plates.

DOI: https://doi.org/10.7554/eLife.37673.005

inducing form of thyroglobulin (Tg$^{cog}$) (*Cameron et al., 2011*). Overall, the gene expression changes detected in all datasets shared some degree of similarity, and 227 genes were commonly changed, showing the activation of ER stress signaling in the MCDS chondrocytes, up-regulation of genes such as *Fgf21* and down-regulation of genes such as *Ldb3*. Differences were also found, and 473 genes were specifically changed in 13del, such as *Apoa4* (up-regulated) and *Atp2a1* (down-regulated) (*Supplementary file 4*). The differences from the published datasets may be due to the different mouse models, the time points analyzed (p14 versus 13del at p10), and also the methods of sampling the various chondrocyte populations. Those transcriptomes were derived from whole proliferative and hypertrophic zones, while ours were generated from precisely fractionated chondrocyte populations from the growth plate.

## PERK-p-eIF2 signaling is the major contributor to chondrocyte adaptation to ER stress

We investigated the contributions of the UPR arms, PERK, IRE1, and ATF6 to the HC response to ER stress. By ontology and pathway analyses of Cluster I, we found enrichment of genes in the PERK pathway and IRE1-Xbp1$^S$ regulated ERAD, but not for ATF6 signaling (*Figure 1D and E*; *Figure 1—figure supplement 1C*; *Supplementary file 2* and *3*). Activation of PERK signaling in 13del HC was demonstrated by up-regulation of p-eIF2$\alpha$ and its downstream components (*Atf4, Atf3, Ddit3, Ero1l* and *Ppp1r15a*) (*Figure 1—figure supplement 2A and C*) which were validated by *in-situ* hybridization and immunostaining (*Figure 1—figure supplement 2B and C*). Using Motif enrichment analysis, we found that the binding motifs of CHOP (encoded by *Ddit3*) and ATF4 were highly enriched in Cluster I, but not those for Xbp1$^S$ or ATF6 (*Figure 1F* and *Supplementary file 5*). By interrogating ATF4 and CHOP ChIP-seq data (*Han et al., 2013*), we found significant over-representation of ATF4 (odds ratio = 2.87, p<0.0001) and CHOP (odds ratio = 4.33, p<0.0001) binding peaks associated with the genes from Cluster I but not for the other clusters (*Supplementary file 6*). Cluster I genes are therefore most likely to be directly regulated by UPR-associated transcription factors.

Together, these data suggest a more prominent contribution of the PERK-p-eIF2 signaling pathway than that of Xbp1$^S$, which is consistent with another MCDS mouse model study that found inactivation of Xbp1 in HCs did not alter the severity of dwarfism (*Cameron et al., 2015b*). To test this notion, we ectopically expressed Xbp1$^S$ in HCs in transgenic mice (*Figure 1—figure supplement 3A*). Overexpression of Xbp1$^S$ specifically in HCs did not affect the growth plate (*Figure 1—figure supplement 3B and C*).

## ATF4 expression in hypertrophic chondrocytes reprogrammes differentiation

Apart from its role in the stress response, ATF4 is also required for chondrocyte differentiation through direct activation of *Ihh* (*Wang et al., 2009*). In the E14.5-E17.5 fetal growth plate, ATF4 is expressed in differentiating chondrocytes, with the highest expression in pHCs (*Figure 2A*). ATF4 expression in HCs progressively decreases after birth and by p10 levels are especially lowered in the LHZ (*Figure 2A*). Therefore, the chondrocyte differentiation defects in the MCDS model might be

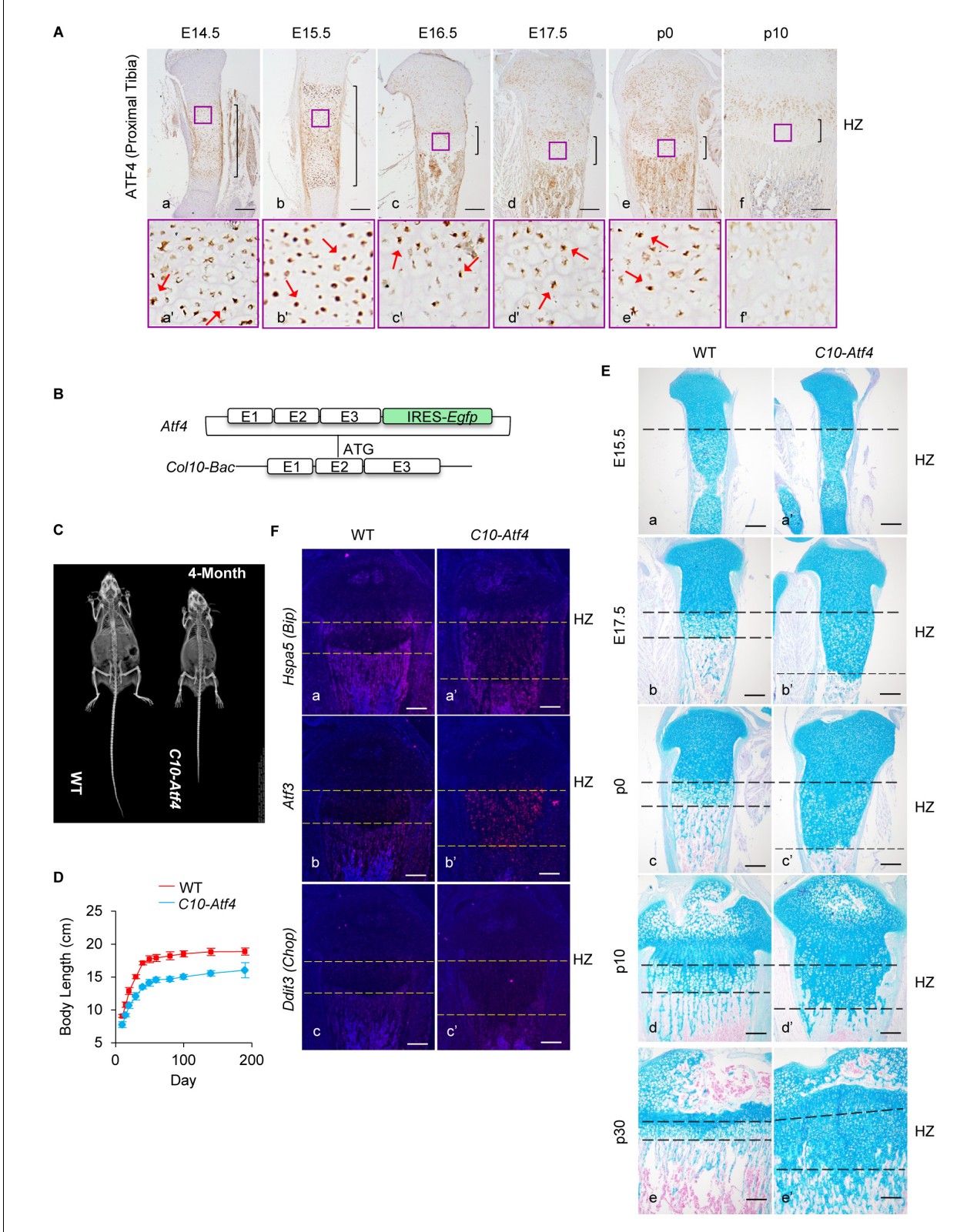

**Figure 2.** Ectopic activation of ATF4 in HCs recaptures the phenotypes of 13del mice. (A) Expression profiles of ATF4 (a–f) on the WT growth plates from E14.5 to P10 stages. Higher magnification of the boxed region (a'–f') was shown to demonstrate the differential expression of ATF4 and the positive cells were arrowed. (Scale Bar = 200 μm) (B) Scheme of *Atf4* expressing vector. *Atf4* cDNA is inserted after the ATG codon in exon 2 of the *Col10a1*-Bac followed by an IRES-*Egfp* cassette. (C) Radiographic analysis revealed the dwarfism and skeletal abnormality of *C10-Atf4* mice at 4-month-
*Figure 2 continued on next page*

*Figure 2 continued*

old stage (n = 3). (D) Body lengths of the WT (n = 9) and *C10-Atf4* (n = 7) littermates were monitored from birth to 30-week stage, and a consistent reduction of body length in *C10-Atf4* mice was observed. (E) Abnormal proximal tibial growth plates with expanded HZ, delimited by dotted lines, were observed in *C10-Atf4* mice by Alcian Blue staining. (Scale bar = 200 μm). (F) Ectopic expression of *Atf4* in HCs was insufficient for ER stress response induction, indicated by *in-situ* hybridization of ER stress markers (*Hspa5, Atf3* and *Ddit3*). (Scale bar = 200 μm).
DOI: https://doi.org/10.7554/eLife.37673.006
The following figure supplement is available for figure 2:

**Figure supplement 1.** *C10-Atf4* transgene is specifically expressed in HCs, and ectopic expression of *Atf4* in HCs does not induce cell apoptosis.
DOI: https://doi.org/10.7554/eLife.37673.007

directly caused by ectopic overexpression of ATF4 in HCs, as a consequence of the preferential translation of *Atf4* transcripts modulated by p-eIF2α.

To assess the impact of ATF4 overexpression in HCs in the absence of ER stress, we generated a transgenic mouse model carrying a *Col10-Bac-Atf4-IRES-Egfp* transgene (hereafter referred to as *C10-Atf4*) (*Figure 2B*), in which ATF4 expression was driven by the highly HC-specific promoter of *Col10a1* (*Leung et al., 2011*; *Yang et al., 2014*). We confirmed HC-specific expression of the *C10-Atf4* transgene in the developing growth plates from fetal (E15.5) to adult (p20) stages (*Figure 2—figure supplement 1A and B*). Similar to 13del mice, adult *C10-Atf4* transgenic mice were dwarfs, being approximately 20% shorter than wild-type littermates (*Figure 2C and D*). Histological analyses revealed growth plate abnormality in both the appendicular and axial skeleton. These defects are illustrated by the greater than three-fold expansion of the HZ of the tibia and vertebrae of *C10-Atf4* mice (*Figure 2E*; *Figure 2—figure supplement 1C and D*). Interestingly, the HZ expansion is more severe in *C10-Atf4* mice (threefold of WT) than that in 13del (2.5-fold of WT), which is paralleling to the expression level of ATF4. Although forced expression of ATF4 in fibroblasts was reported to decrease survival (*Han et al., 2013*), cell viability was not affected in *C10-Atf4* mice (*Figure 2—figure supplement 1E*). Importantly, overexpression of ATF4 in HCs in the absence of ER stress did not induce transcription of the UPR-associated genes *Hspa5* and *Ddit3* (*Figure 2F*), although *Atf3* was slightly upregulated (*Han et al., 2013*). Therefore, activation of ATF4 alone, in the absence of the ER stress response, is sufficient to alter HC differentiation, disturb endochondral ossification and cause skeletal abnormalities similar to those observed in 13del mice.

## ATF4 reprograms chondrocyte hypertrophy by directly activating *Sox9*

In *C10-Atf4* HCs, constitutive ATF4 activation down-regulated the expression of *Col10a1* and led to persistent expression of prehypertrophic chondrocyte marker genes *Sox9, Col2a1, Ppr* and *Ihh* in the HZ (*Figure 3A*). However, BrdU-labeled HCs were not detectable in the *C10-Atf4* HZ after 2 hr (*Figure 2—figure supplement 1F*), suggesting *C10-Atf4* HCs did not appear to have progressed through the $G_1$/S checkpoint. The sequential differentiation process in growth plate chondrocytes is tightly regulated by multiple chondrocyte-specific transcription factors that control the expression of cell type-specific genes and secreted growth factors (*Hojo et al., 2016*; *Leung et al., 2011*; *Stricker et al., 2002*; *Koziel et al., 2005*; *Ionescu et al., 2012*; *Akiyama et al., 2002*; *Liu et al., 2017*). We searched the published ER stress-associated ATF4 ChIP-Seq data (*Han et al., 2013*) for binding peaks in crucial chondrogenic transcription factor genes, including members of SOX, RUNX, MEF2, GLI and FOXA families. We found ATF4 binding peaks in the regulatory regions of *Sox9, Sox5, Sox6, Runx2, Gli2* and *Gli3*. Amongst these, only the *Sox* genes were up-regulated in 13del middle and lower zones, but not *Gli2* or *Gli3* (*Figure 3—figure supplement 1A and B*), suggesting that the *Sox* family could be the targets of ATF4. Notably, overexpression of *Sox9* in HCs resulted in an expansion of HZ and impaired terminal differentiation of HCs, similar to the phenotypes observed in *C10-Atf4* mice (*Hattori et al., 2010*). Moreover, the expression pattern of SOX9 paralleled that of ATF4 in 13del and *C10-Atf4* mice, raising the possibility of a direct interaction between these two factors as part of the molecular mechanism underlying the MCDS pathology.

SOX9 is highly expressed in immature chondrocytes, transactivates critical cartilaginous matrix genes and regulates chondrocyte proliferation, differentiation and entry into hypertrophy (*Leung et al., 2011*; *Akiyama et al., 2002*; *Liu et al., 2017*; *Bell et al., 1997*; *Dy et al., 2012*). It is required for expression of SOX5 and SOX6, which cooperate with SOX9 to transactivate *Col2a1* (*Lefebvre et al., 1998*; *Liu and Lefebvre, 2015*). We identified two putative C/EBP-ATF4

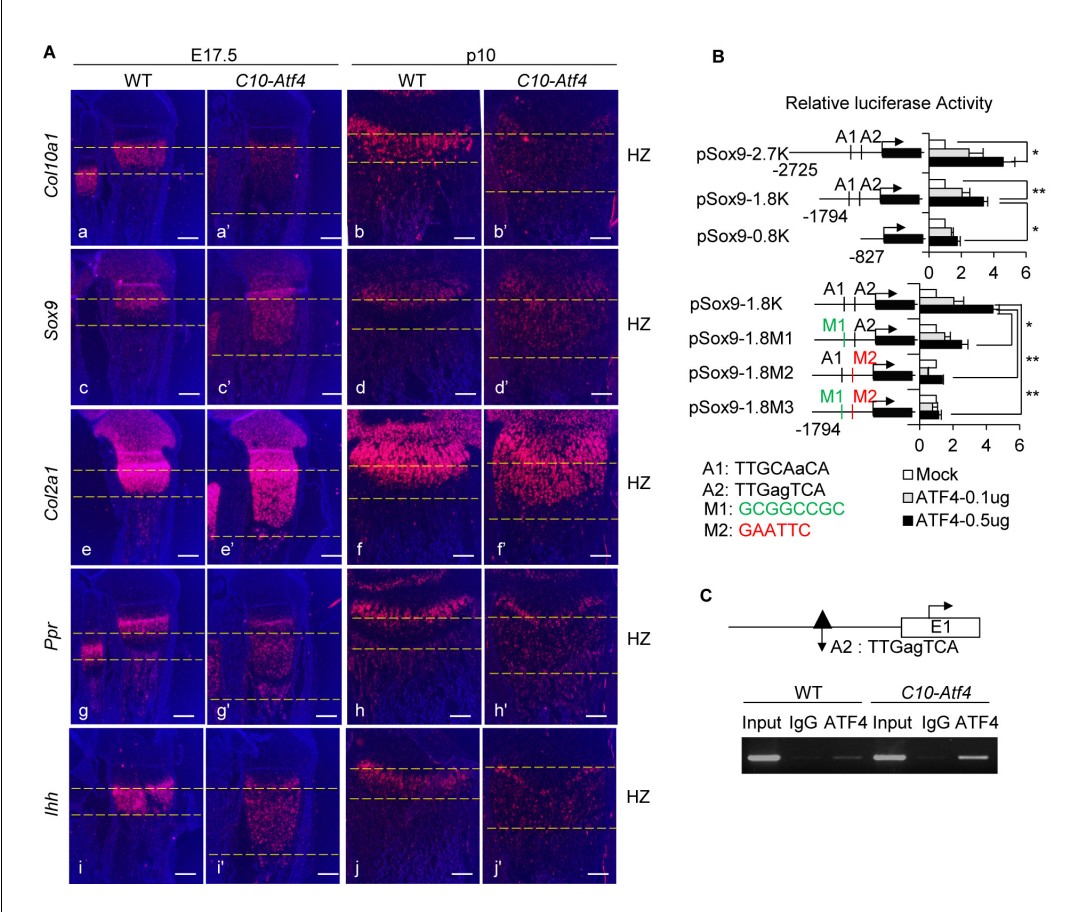

**Figure 3.** ATF4 governs de-differentiation of HCs via direct regulation of *Sox9*. (**A**) Ectopic expression of *Atf4* in HCs leads to accumulation of premature chondrocytes in *C10-Atf4* HZ, indicated by expression patterns of chondrogenic markers *Col10a1* (**a, a', b, b'**), *Sox9* (**c, c', d, d'**), *Col2a1* (**e, e', f, f'**), *Ppr* (**g, g', h, h'**) and *Ihh* (**i, i', j, j'**) (Scale bar = 200 μm). (**B**) Luciferase activities of reporters driven by *Sox9* promoter with different lengths (pSox9-2.7K, pSox9-1.8K, and pSox9-0.8K) (i) or ATF4 putative binding sites mutants (pSox9-1.8M1, pSox9-1.8M2 and pSox9-1.8M3) (ii) responding to different dosages of ATF4 were measured in ATDC5 cells and results were presented as fold induction compared with mock-transfected cells from three independent experiments. Error bars were shown as S.D. and significance was determined by unpaired Two-tailed Student's *t*-test. *: p-value<0.05, **: p-value<0.005, ***: p-value<0.0005. (**C**) ChIP-PCR showed the direct binding of ATF4 to the putative motif on the *Sox9* promoter in vivo, using the nuclear extracts from E15.5 WT and *C10-Atf4* limb chondrocytes. Three independent experiments were performed and one presentative result was shown. An ATF4 ChIP-seq peak (dark triangle) around this region has been identified in ER-stressed MEF cells.

DOI: https://doi.org/10.7554/eLife.37673.008

The following figure supplement is available for figure 3:

**Figure supplement 1.** Presentation of ATF4 ChIP peaks on regulatory region of vital chondrogenic transcriptional factors.
DOI: https://doi.org/10.7554/eLife.37673.009

motifs, named A1 and A2, in the ATF4-binding peak in the *Sox9* promoter region. By transfection assays in ATDC5 chondrocyte cells, we found ATF4 could transactivate luciferase reporters controlled by the *Sox9* promoter (*Figure 3B*). Mutation of A1 and A2 respectively reduced or abolished ATF4 activation of the *Sox9* reporters (*Figure 3B*). Anti-ATF4 ChIP-PCR assays, using nuclear extracts from E15.5 wild-type and *C10-Atf4* limbs, demonstrated that ATF4 binds directly to the putative motif region on the *Sox9* promoter in vivo (*Figure 3C*).

We next assessed the contribution of ATF4 activation of *Sox9* in reverting HC differentiation by conditionally inactivating *Sox9* in *C10-Atf4* HC, using HC-specific Col10a1-Cre (*Yang et al., 2014*) (*Figure 4—figure supplement 1A*). In the absence of *Sox9*, the expansion of HZ in *C10-Atf4* mice was markedly reduced, and there were fewer cells expressing *Col2a1* in the HZ (*Figure 4A–C*). Moreover, conditional inactivation of *Sox9* in 13del mice decreased expression of *Col2a1* and *Ppr* in HC, and the HZ expansion was considerably shortened (*Figure 4D–F*; *Figure 4—figure supplement*

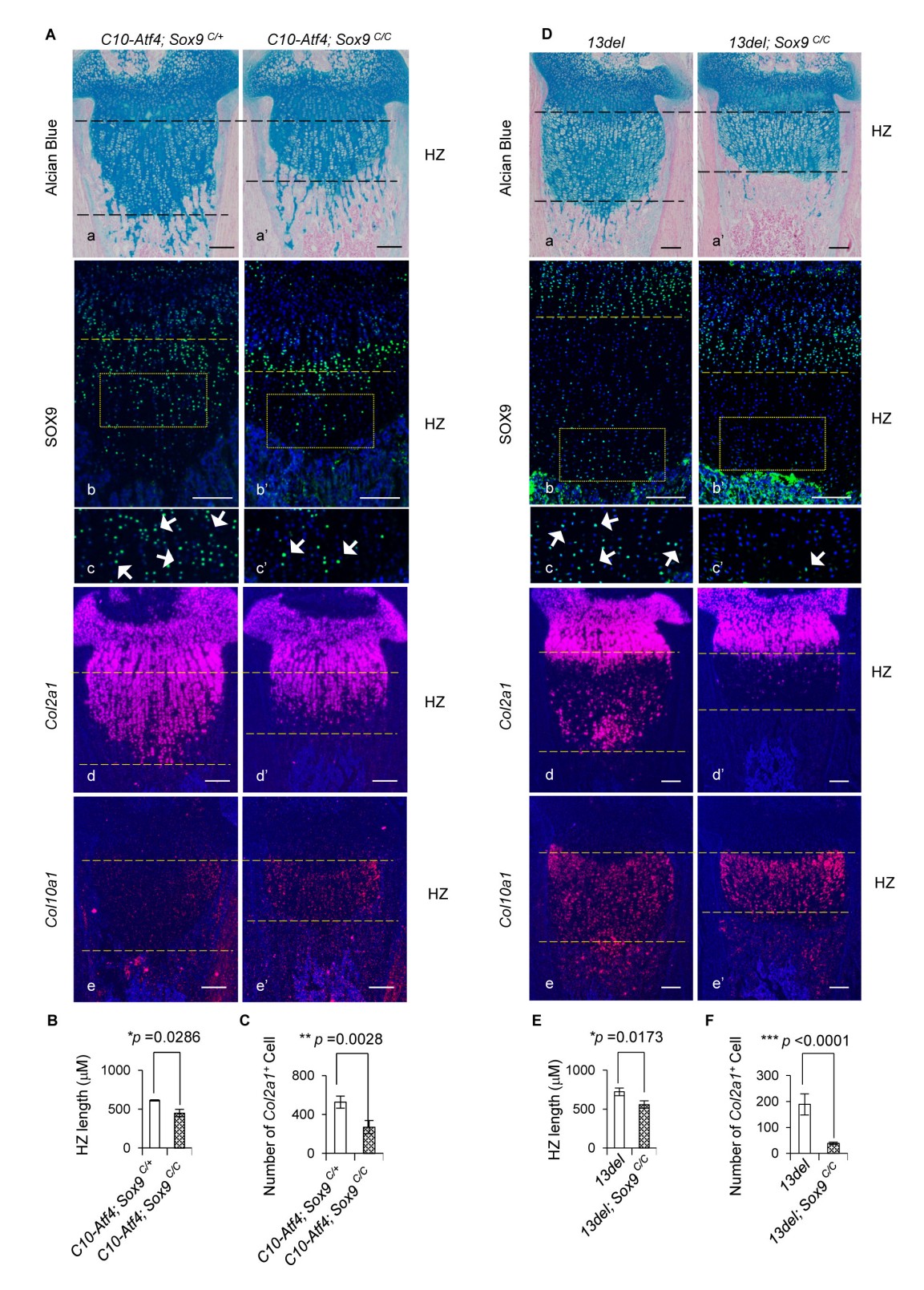

**Figure 4.** Genetic rescue of growth plate abnormalities in *C10-Atf4* and 13del mice via HC-specific inactivation of *Sox9*. (**A**) Removal of *Sox9* in *C10-Atf4* HCs rescued growth plate abnormalities of *C10-Atf4* mice at p10 stage, shown by histology (**a, a'**), expression analyses of SOX9 (**b, b', c, c'**), *Col2a1* (**d, d'**) and *Col10a1* (**e, e'**). Higher magnification of the boxed region was shown to demonstrate the differential expression of SOX9 and the positive cells were indicated by arrows. (Scale bar = 200 µm). (**B**) Measurement of the HZ lengths of *C10-Atf4* and *C10-Atf4;Sox9*[C/C] littermates (n = 5).

*Figure 4 continued on next page*

*Figure 4 continued*
The vertical length of the central part of the HZ from 10 different sections of each mouse was averaged. (C) Quantification of *Col2a1* positive cells in HZ of *C10-Atf4* and *C10-Atf4; Sox9^{C/C}* littermates (n = 5). For each mouse, the number of positive cells was counted and average on five non-adjacent sections. (D) Removal of *Sox9* in 13del HCs rescued growth plate abnormalities of 13del mice at p10 stage (n = 5), shown by histology (a, a'), expression analyses of SOX9 (b, b', c', c'), *Col2a1* (d, d') and *Col10a1* (e, e'). Higher magnification of the boxed region was shown to demonstrate the differential expression of SOX9 and the positive cells were arrowed. (Scale Bar = 200 μm). (E) Measurement of the HZ lengths of 13del and 13del;*Sox9^{C/C}* littermates (n = 5). The vertical length of the central part of the HZ from 10 different sections of each mouse was averaged. (F) Quantification of *Col2a1* positive cells in HZ of 13del and 13del;*Sox9^{C/C}* littermates (n = 5). For each mouse, the number of positive cells was counted and average on five non-adjacent sections. Error bars were shown as S.D. and significance was determined by Two-tailed Mann-Whitney *U*-test. *: p-value<0.05, **: p-value<0.005, ***: p-value<0.0005.
DOI: https://doi.org/10.7554/eLife.37673.010
The following figure supplement is available for figure 4:

**Figure supplement 1.** Removal of SOX9 in HCs does not cause any abnormalities in growth plates in WT mice but reduces the number of immature HCs in 13del mice.
DOI: https://doi.org/10.7554/eLife.37673.011

*1B and C*). Deletion of *Sox9* in wild-type HCs did not affect chondrocyte hypertrophy (*Figure 4—figure supplement 1D*). Collectively, these data suggest ER stress-induced overexpression of ATF4 reverts differentiation in 13del HC by direct activation of *Sox9* in HCs, thereby perturbing chondrocyte hypertrophy.

## CHOP plays an adaptive and pro-survival role in 13del HC

CHOP is another prominent transcription factor that was active in 13del HCs, revealed by bioinformatics analysis. It is preferentially expressed in the PERK signaling pathway, downstream of p-eIF2α and ATF4, which regulates protein synthesis via the PPP1R15A negative feedback loop and restores protein synthesis and induces oxidative stress via *Ero1l* (*Han et al., 2013*; *Marciniak et al., 2004*). Apart from that, overexpression of CHOP in the bone microenvironment in transgenic mice has been reported to impair osteoblastic function leading to osteopenia (*Pereira et al., 2007*), while CHOP null mice show retarded bone formation (*Pereira et al., 2006*), indicating its role in regulating osteoblast differentiation. Although CHOP is widely considered as a pro-apoptotic factor, it has context- and cell-type specific roles as an adaptive and pro-survival factor in several diseases (*Pennuto et al., 2008*; *Southwood et al., 2002*; *Lu et al., 2014*; *Moreno et al., 2012*). Forced expression of ATF4 and CHOP has been reported to increase cell death (*Hartley et al., 2013*). We, therefore, assessed the contribution of CHOP in the adaptation of 13del HCs.

We found ablating CHOP encoding gene *Ddit3* in 13del mice exacerbated the skeletal defects and growth plate phenotype. The 13del;*Ddit3^{-/-}* mice displayed further tibial shortening (*Figure 5A and B*) with more significant (~20%) HZ expansion (*Figure 5C and D*), and increased the number of chondrocytes expressing immature chondrogenic markers SOX9, *Col2a1* and *Ppr* in the HZ (*Figure 5C and E*). Strikingly, in contrast to 13del, there was increased apoptosis in 13del;*Ddit3^{-/-}* HC, consistent with a pro-survival role for CHOP (*Figure 5F and G*).

Our transcriptome analyses of fractionated 13del;*Ddit3^{-/-}* growth plates revealed upregulation of molecular chaperones (*Hspa5, Dnajb9, Dnajb11* and *Canx*) and ER stress sensors (*Xbp1* and *Atf4*) in the MHZ and LHZ (*Figure 5—figure supplement 1B*), which further support the positive correlation between the phenotype severity and the expression level of ATF4. In contrast to the elevated stress level, the PERK signaling pathway was enfeebled, reflected by marked (>3.5 fold) down-regulation of CHOP targets *Atf3, Ppp1r15a* and *Ero1l*, indicating the phenotype severity reflected by aberrant cell differentiation is probably independent of those downstream factors. These results are in contrast to the pro-apoptotic role reported for CHOP in a mouse model of Pseudoachondroplasia caused by expression of misfolded COMP in proliferating and hypertrophic chondrocytes, where deleting CHOP reduced apoptosis but exacerbated growth plate chondrocyte disorganisation (*Posey et al., 2012*; *Piróg et al., 2014*). These differences may be due to variation in the responses of proliferating versus hypertrophic chondrocytes and/or the acuteness and duration of the ER stress. Therefore, CHOP aids in the cell adaptation to stress and mediates survival in13del HC, and it is important to identify pro-survival/anti apoptotic factor(s) downstream of CHOP.

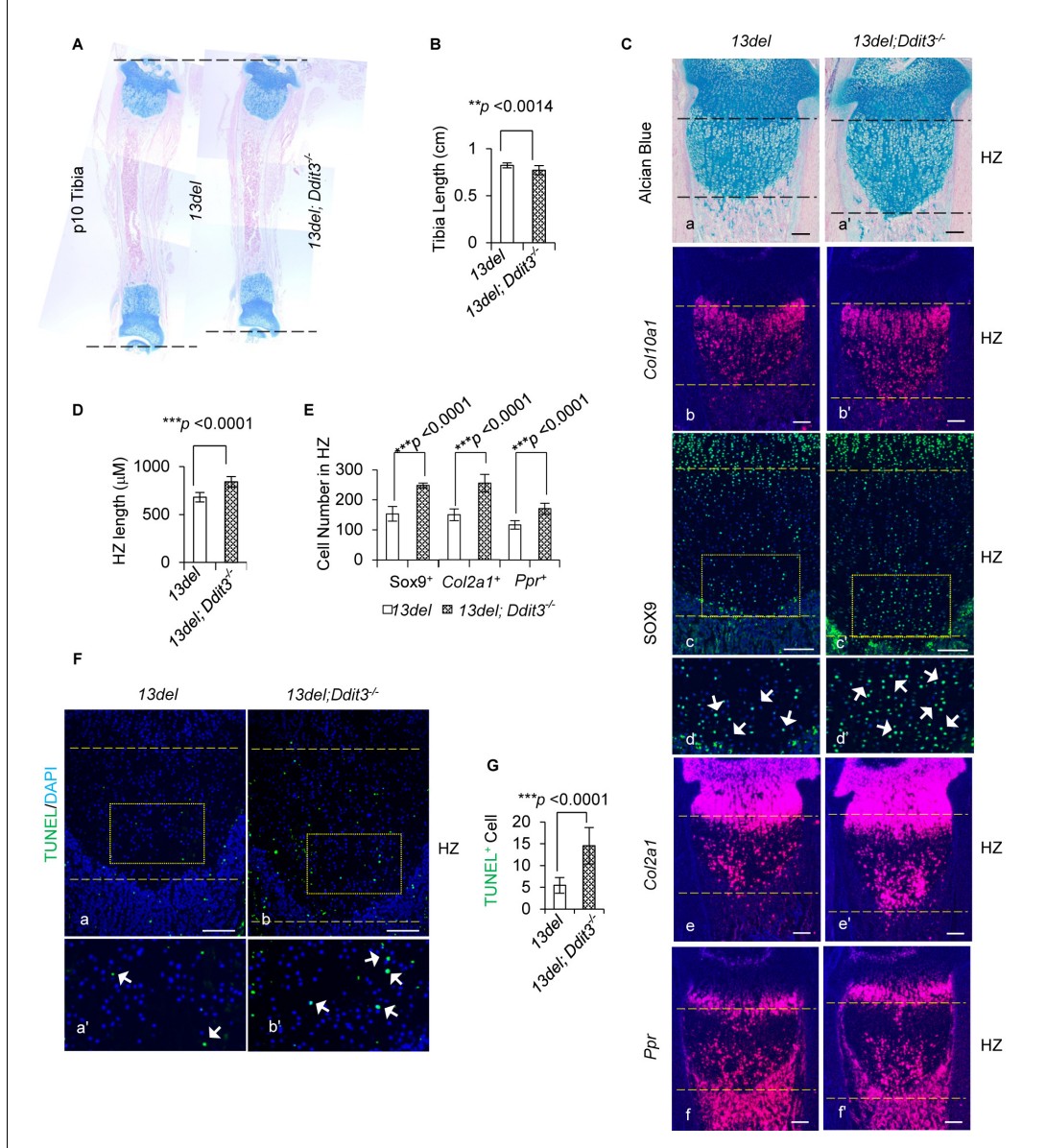

**Figure 5.** CHOP limits the deleterious consequence of ER Stress in 13del mice. (**A–B**) The tibia length is further shortened in 13del;*Ddit3*[-/-] mice at the p10 stage. The comparison was performed between 13del and 13del;*Ddit3*[-/-] littermates (n = 5). (**C**) Exacerbated growth plate abnormalities were observed in 13del mice with global loss of *Ddit3* at p10 stage (n = 5), shown by histology (**a, a'**), expression analyses of *Col10a1* (**b, b'**), SOX9 (**c, c', d, d'**), *Col2a1* (**e, e'**) and *Ppr* (**f, f'**). Higher magnification of the boxed region was shown to demonstrate the differential expression of SOX9 and the positive cells were indicated by arrows. (Scale Bar = 200 μm) (**D**) Measurement of the HZ lengths of 13del and 13del;*Ddit3*[-/-] littermates (n = 5). The vertical length of the central part of the HZ from 10 different sections of each mouse was averaged. (**E**) Quantification of SOX9, *Col2a1* and *Ppr*-positive cells in HZ of 13del and 13del;*Ddit3*[-/-] littermates (n = 5). For each mouse, the number of positive cells was counted and average on five non-adjacent sections. (**F–G**) TUNEL assay revealed an increased number of apoptotic cells in 13del;*Ddit3*[-/-] HZ (n = 5). Arrows indicate TUNEL-positive cells. (Scale Bar = 200 μm). For each mouse, the number of positive cells was counted and average on five non-adjacent sections. Error bars were shown as S.D. and significance was determined by Two-tailed Mann-Whitney *U*-test. *: p-value<0.05, **: p-value<0.005, ***: p-value<0.0005.

DOI: https://doi.org/10.7554/eLife.37673.012

The following figure supplement is available for figure 5:

**Figure supplement 1.** Impaired PERK signaling pathway results in elevated ER stress level in 13del;*Ddit3*[-/-] mice.

DOI: https://doi.org/10.7554/eLife.37673.013

## ATF4 and CHOP mediate chondrocyte survival by activating *Fgf21*

CHOP acts, not only downstream of ATF4 but also as its interacting partner in modulating ER stress targets (*Han et al., 2013*). To elucidate the pro-survival role of the PERK signaling pathway in 13del HC, we searched for target genes of CHOP and ATF4 in Cluster I (*Supplementary file 6*). We found *Fgf21*, a reported target of ATF4 (65), was the most upregulated gene in 13del HCs (*Figure 6—figure supplement 1A* and *Supplementary file 1*), which was confirmed by *in-situ* hybridization and immunoblotting (*Figure 6A*). *Fgf21* has been reported to be similarly activated in ER-stressed chondrocytes (*Cameron et al., 2011*).

FGF21 is a hormone with roles in glucose and lipid metabolism (*Kharitonenkov et al., 2005*) and plays a survival role in response to diverse stressful conditions, such as amino acid deprivation, mitochondrial stress and ER stress-associated diseases such as diabetes, cardiovascular diseases (reviewed in [*Gómez-Sámano et al., 2017*; *Kim and Lee, 2015*; *Salminen et al., 2017*]). We found *Fgf21* expression was effectively turned on (>100 fold increased expression) in response to

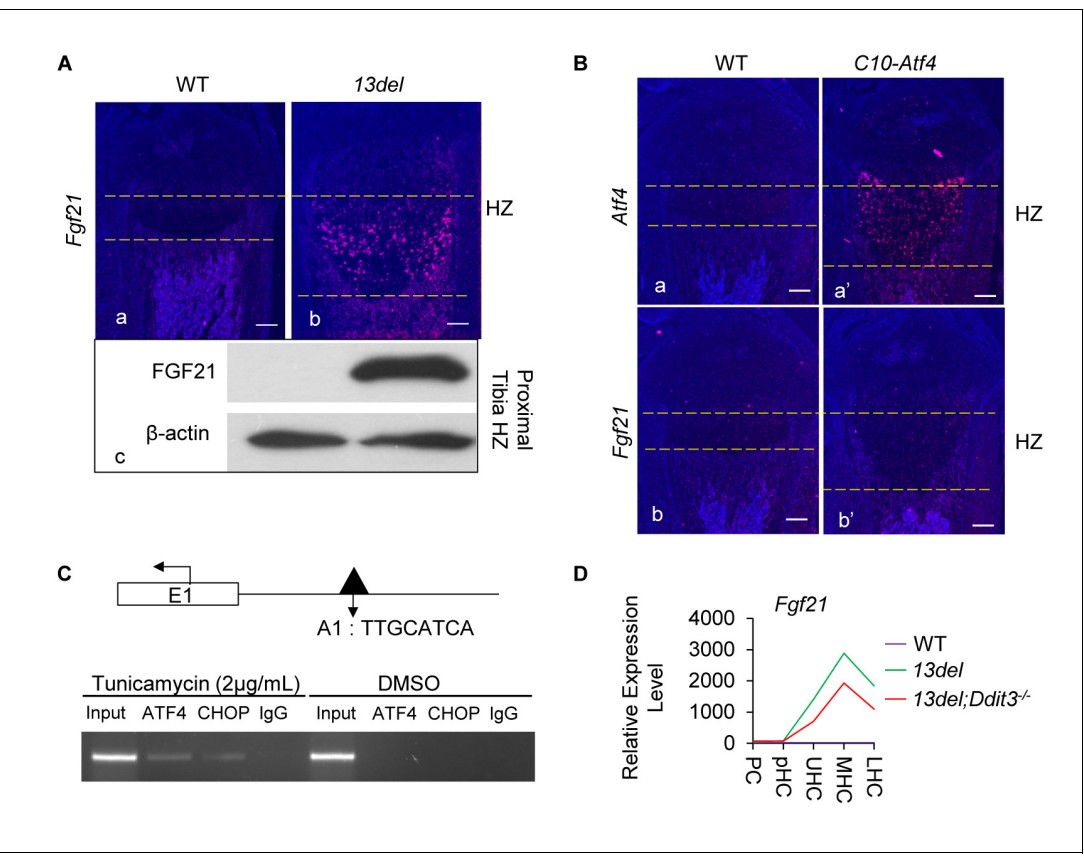

**Figure 6.** *Fgf21* is regulated by ATF4 and CHOP. (**A**) Significant activation of *Fgf21* in 13del HCs at the p10 stage, revealed by *in-situ* hybridization (**a, b**) and western blot (**c**) (Scale Bar = 200 μm). (**B**) Ectopic expression of *Atf4* (**a, a'**) is insufficient for *Fgf21* (**b, b'**) induction in HCs (Scale Bar = 200 μm). (**C**) ChIP-PCR showed the binding of ATF4 and CHOP to the putative motif on the *Fgf21* promoter under ER stress in NIH3T3 cells. (**D**) Normalized microarray measurement of *Fgf21* in WT, 13del and 13del;*Ddit3*$^{-/-}$ chondrocytes. (**E–F**) FGF21 protects the 13del HCs from apoptosis in a dosage-dependent manner. Arrows indicate TUNEL-positive cells. The quantification of TUNEL positive cells was performed between littermates (n = 5). For each mouse, the number of positive cells was counted and averaged on five non-adjacent sections. (Scale Bar = 200 μm) Error bars were shown as S.D. and significance was determined by Two-tailed Mann-Whitney *U*-test. *: p-value<0.05, **: p-value<0.005, ***: p-value<0.0005.

DOI: https://doi.org/10.7554/eLife.37673.014

The following figure supplement is available for figure 6:

**Figure supplement 1.** FGF21 was activated by ER stress via ATF4 and CHOP direct regulation.
DOI: https://doi.org/10.7554/eLife.37673.015

treatment with the ER stress inducer tunicamycin in fibroblasts (NIH3T3 and MEF cells) and ATDC5 cells (*Figure 6—figure supplement 1B and C*).

We next tested the functional relevance of a reported C/EBP-ATF4 binding motif in the *Fgf21* promoter (*Han et al., 2013*) that coincided with an ATF4 peak. By transactivation assays (*Figure 6—figure supplement 1D*), we found deleting (pFgf21-Luc3 and pFgf21-Luc4) or mutating (pFgf21-M1 and pFgf21-M3) the ATF4 binding motif abolished the activation induced by ER stress. However, *Fgf21* was not induced in *C10-Atf4* mice (*Figure 6B*), suggesting ATF4 is necessary but not sufficient for *Fgf21* induction. The need for another factor is supported by the ChIP assays where we found both ATF4 and CHOP bind to the ATF4 motif-containing peak region in the cells under ER stress (*Figure 6C*), even though no CHOP binding peak was detected in the *Fgf21* promoter. Expression of *Fgf21* was down-regulated by approximately 40% in 13del;*Ddit3*$^{-/-}$ HC despite the upregulation of *Atf4* (*Figure 6D*), consistent with the requirement of ATF4-CHOP cooperation for *Fgf21* induction under ER stress.

We assessed whether FGF21 had a survival role in 13del HC by genetically ablating the gene (*Figure 7—figure supplement 1A*). *Fgf21* null mice have normal growth plates and HC viability (*Figure 7—figure supplement 1B and C*). The HZ expansion in 13del;*Fgf21*$^{-/-}$ mice was comparable to that of 13del mice, and the reverted differentiation process was not affected (*Figure 7—figure supplement 1D–F*). However, we found increased apoptosis in the HZ of *Fgf21*-deficient 13del mice, and this protective effect of FGF21 is dosage dependent (*Figure 7Aand B*).

To determine whether the pro-survival role of FGF21 was cell autonomous or non-cell autonomous in 13del HCs, we tested the survival of 13del HCs carrying an *Fgf21* null mutation, in mouse chimeras. We utilised compound mutants carrying the 13del transgene and a *Col10a1*$^{Egfp}$ allele [*Egfp* knocked into the *Col10a1* gene (*Yang et al., 2014*)] so that all 13del HCs are marked by GFP expression. We created mouse chimeras by aggregating 13del;*Col10a1*$^{Egfp}$ and 13del;*Fgf21*$^{-/-}$ morulae (*Figure 7C*). In the ensuing chimeras, 13del HCs are marked by EGFP expression, and express *Fgf21*/FGF21. Mice with different degrees of 13del;*Col10a1*$^{Egfp}$/13 del;*Fgf21*$^{-/-}$ chimerism, were analyzed for HC survival. Similarly as found with 13del;*Fgf21*$^{-/-}$ compound mutants, in 13del; *Col10a1*$^{EGFP}$/13 del;*Fgf21*$^{-/-}$ chimeras, more 13del;*Fgf21*$^{-/-}$ HCs (non-EGFP$^+$ HC population) underwent apoptosis, than the 13del; *Col10a1*$^{Egfp}$ HCs (GFP$^+$ HC population) (*Figure 7D*). Moreover, there was a positive correlation between the contribution of13del;*Col10a1*$^{Egfp}$ HCs and the number of surviving cells in the HZ (*Figure 7E*), consistent with a protective role for FGF21 in the ER-stressed 13del HCs. Furthermore, we found that *13del;Fgf21*$^{-/-}$ HCs adjacent to *13del;Col10a1*$^{Egfp}$ HCs expressing *Fgf21* still underwent apoptosis (*Figure 7F*). This inability of *13del;Col10a1*$^{Egfp}$ to rescue *13del;Fgf21*$^{-/-}$ HCs suggests that FGF21 protects HCs from apoptosis, cell autonomously.

## ISRIB, an ISR p-eIF2α signaling inhibitor, can ameliorate 13del skeletal deformities

Upon ER stress, PERK phosphorylation of eIF2α is the critical upstream controlling point that triggers the p-eIF2α/ATF4/CHOP signaling pathway (*Pakos-Zebrucka et al., 2016*). Our data show that genetically ablating the essential transcription factor CHOP in the p-eIF2α signaling pathway as a strategy for rescuing the aberrant chondrocyte differentiation is imperfect, because of effects on cell survival and stress aggravation. Also, addressing the impact of transcription factor overexpression and cell-type specificity is required because ATF4 is essential for normal development. Therefore, it is necessary to identify a suitable entry point in the pathway which can be manipulated for protection or rescue from the deleterious effects of ER stress, without interfering with normal developmental function.

Recently, a small molecule, Integrated Stress Response InhiBitor (ISRIB) has been reported to be a potent ISR signaling inhibitor, rendering cells insensitive to eIF2α phosphorylation by targeting the interaction between eIF2 and eIF2B, and its activity is independent of eIF2a phosphorylation (*Sekine et al., 2015*; *Sidrauski et al., 2015*). ISRIB shows acceptable pharmacokinetic properties and no overall toxicity in mice and has been reported to show significant neurotrophic effects in mice (*Sekine et al., 2015*; *Di Prisco et al., 2014*).

We tested the potential of ISRIB to modify the chondrodysplasia phenotype by treating 13del and wild-type littermates with ISRIB (2.5 mg/kg) or vehicle twice daily by intraperitoneal injection from E13.5 (onset of expression of 13del) to postnatal day 20 (p20) (*Figure 8A*). In wild-type mice, ISRIB had no adverse effects on weight gain or body growth (*Figure 8—figure supplement 1A and*

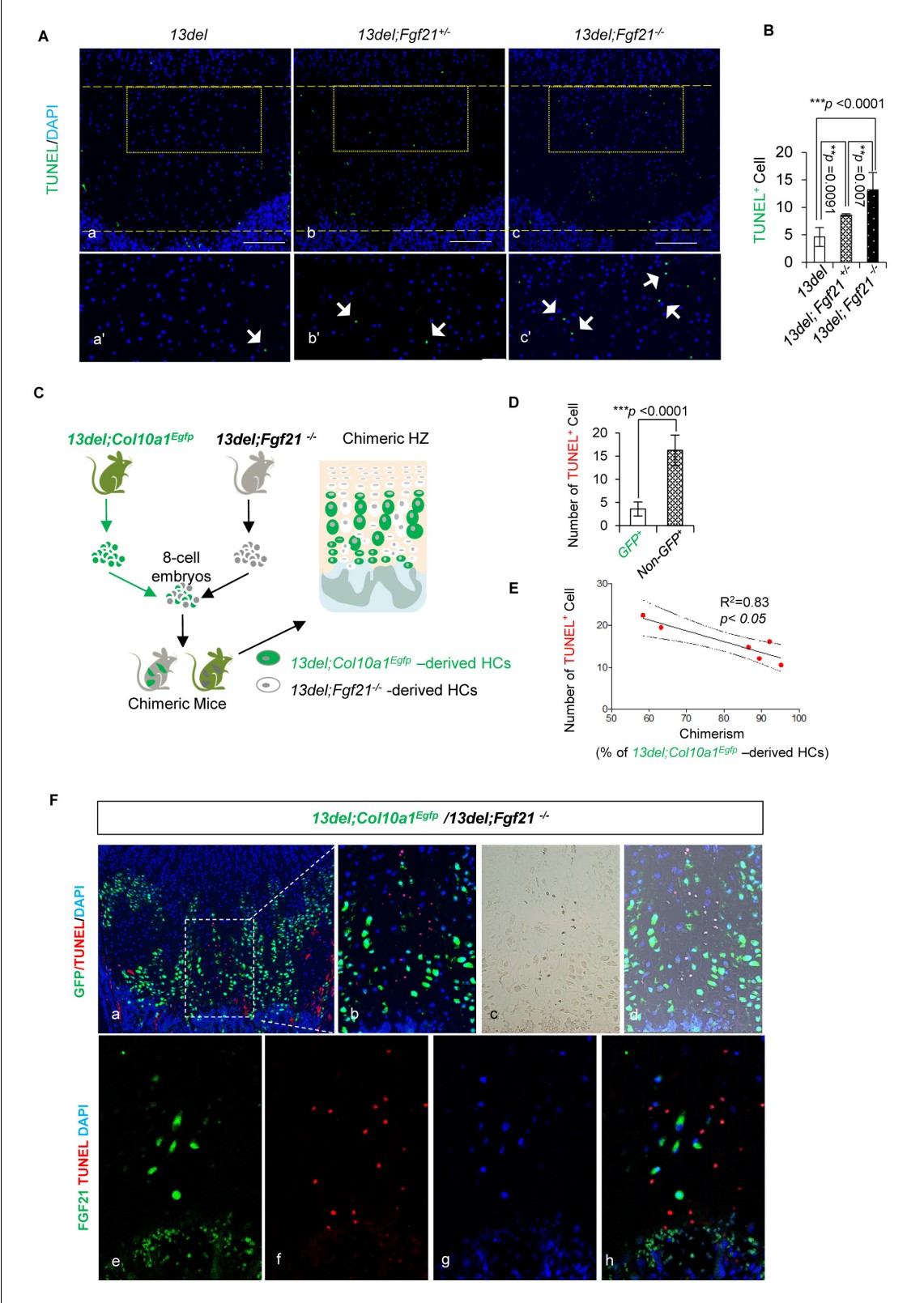

**Figure 7.** FGF21 protects the HCs from ER stress-induced apoptosis in a cell autonomous manner. (A–B) FGF21 protects the 13del HCs from apoptosis in a dosage-dependent manner. Arrows indicate TUNEL positive cells. The quantification of TUNEL-positive cells was performed between littermates (n = 5). For each mouse, the number of positive cells was counted and averaged on five non-adjacent sections (Scale Bar = 200 μm). (C) Schematic diagram of 13del;*Col10a1*<sup>Egfp</sup>/13del;*Fgf21*<sup>-/-</sup> chimera generation. In chimeric HZ, 13del HCs expressing *Fgf21* (13del;*Col10a1*<sup>Egfp</sup>) can be

*Figure 7 continued on next page*

*Figure 7 continued*
distinguished from 13del;*Fgf21*[-/-] HCs by EGFP expression. (D) The number of TUNEL-positive cells was quantified in GFP[+] and GFP negative HCs (n = 7). Error bars were shown as S.D. and significance was determined by Two-tailed Mann-Whitney *U*-test. *: p-value<0.05, **: p-value<0.005, ***: p-value<0.0005. (E) The correlation between cell death and chimerism (indicated by the percentage of 13del-EGFP expressing cells in HZ) in 13del; *Col10a1*[Egfp]/13del;*Fgf21*[-/-] chimeras. (F) Boxed region (a) is shown to demonstrate the differential localization of EGFP and TUNEL signal in chimeric HZ. Higher magnification is shown in b-d. Immunostaining for FGF21 and TUNEL assays (e–h) were applied on the chimeric growth plate, showing the mutually exclusive location of FGF21 (green) and TUNEL signal (red).
DOI: https://doi.org/10.7554/eLife.37673.016
The following figure supplement is available for figure 7:

**Figure supplement 1.** Removal of Fgf21 does not affect the reverted differentiation of 13del HCs.
DOI: https://doi.org/10.7554/eLife.37673.017

*B*). However, ISRIB markedly reduced the dwarfism of 13del mice from newborn to juvenile stages comparing to control group (*Figure 8—figure supplement 1C*). Radiographic analyses revealed treatment with ISRIB ameliorated the skeletal deformities at p20 (*Figure 8B and C*), including the length of tibia/femur and spine; tibia bowing (*genu varum*: the angle between proximal head and distal head of tibia); pelvic bone orientation (the angle between ilium and pubis), and *coxa vara* (narrowed angle between the proximal head and the shaft of the femur) (*Figure 8B*).

We found the HZ expansion in the limb growth plates of ISRIB-treated 13del mice was significantly reduced, and the numbers of SOX9[+], *Col2a1*[+] and *Ppr*[+] cells in the HZs at p10 and p20 were diminished (*Figure 8D–G*). ISRIB with indicated dosage had no observable effect on the limb growth plates in wild-type mice (*Figure 8—figure supplement 1D*). ISRIB treatment in 13del mice also reduced the deformities in other growth plates such as in the axial skeleton, with reduced HZ expansion and decreased the number of *Sox9*[+] and *Col2a1*[+] premature cells in tail intervertebral disc growth plates (*Figure 8—figure supplement 2A–C*).

Furthermore, the effectiveness of postnatal treatment of ISRIB in 13del MCDS was estimated by treating 13del and wild-type littermates with ISRIB (2.5 mg/kg or 5 mg/kg) or vehicle twice daily by intraperitoneal injection from newborn (p0) to 4-week stages (*Figure 8—figure supplement 3A*). Similarly, ISRIB ameliorated the dwarfism of 13del mice from newborn to 4-week stages in a dosage-dependent manner (*Figure 8—figure supplement 3B*), and ameliorated the skeletal deformities (*Figure 8—figure supplement 3C and D*), including the length of tibia/femur and spine; tibia bowing; pelvic bone orientation and *coxa vara*, revealed by radiographic analyses. Thus, without any apparent adverse effect, ISRIB corrected the molecular, histological, and skeletal defects in 13del mice.

## The PERK-p-eIF2α signaling pathway is downregulated explicitly by treatment of ISRIB

The impact of ISRIB on p-eIF2α signaling induced by ER stress in 13del HCs was further addressed. Consistent with the previous study (*Sidrauski et al., 2013*) that the activity of ISRIB is independent of eIF2α phosphorylation (*Sekine et al., 2015*; *Sidrauski et al., 2015*), the expression of p-eIF2α was not affected in ISRIB-treated 13del HCs (*Figure 9A*). Given the fact that translational regulation of ATF4 by p-eIF2α is central to the activation of PERK signaling, the protein expression level of ATF4 in p10 vehicle-treated WT, vehicle- and ISRIB-treated 13del HCs were examined via immunoblotting. As expected, ISRIB evidently inhibited the preferential translation of ATF4 in ER-stressed 13del HCs (*Figure 9B*). The proximal growth plates from above mentioned groups of mice were further fractionated into sub-populations (PC, pHC, UHC, MHC, and LHC), and the transcriptional expression levels of major components of p-eIF2α signaling (*Atf4, Atf3, Ddit3* and *Fgf21*), *Xbp1*[S] and *Hspa5* (*Bip*) were examined by qRT-PCR. Consistently, ISRIB treatment significantly downregulated the mRNA expression level of *Atf3* and *Ddit3* in 13del HCs, indicating the downstream signaling cascade of ATF4 were inhibited (*Figure 9C*), although the expression level of *Atf4* was not affected. The expression patterns and levels of examined factors involved in the p-eIF2α signaling pathway were further validated via *in-situ* hybridization and immunostaining (*Figure 9—figure supplement 1A and B*). Notably, the expression of CHOP and FGF21 was significantly lowered but still detectable. On the other hand, the expression level of total *Xbp1* (*Xbp1*[T]) and spliced *Xbp1* (*Xbp1*[S]) in ISRIB-treated 13del HCs was comparable to vehicle-treated HCs, indicating IRE1/Xbp1 signaling

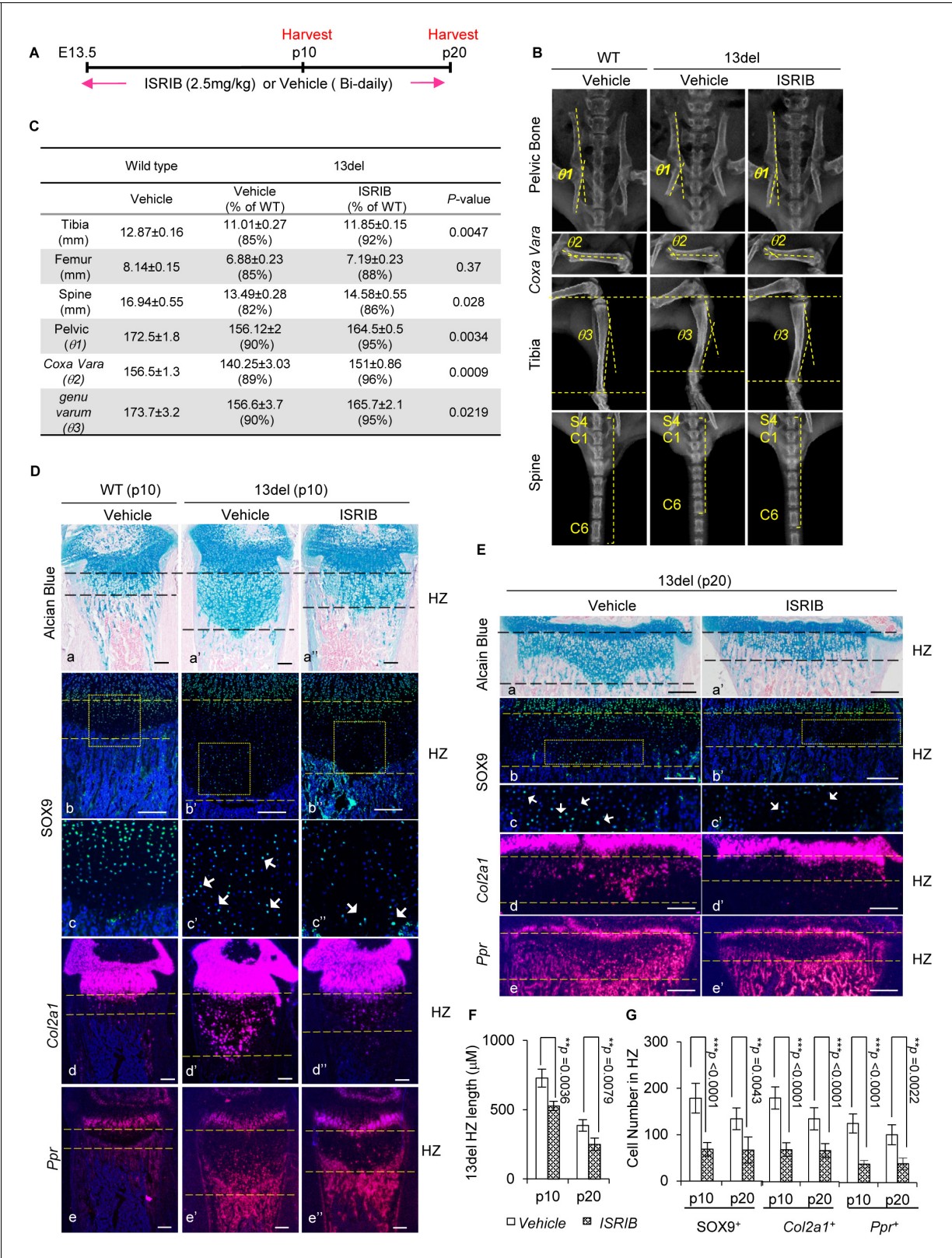

Figure 8. The small molecule ISRIB, preventing ATF4 induction under ER stress, ameliorates *13del* skeletal deformities. (A) Schematic timeline of the ISRIB (2.5 mg/kg) or vehicle (0.5% DMSO in 0.9% saline) administration. The mice were administrated by intraperitoneal injection, starting from E13.5 to the p20 stage. The animals were harvested at indicated time-points. (B–C) Radiographic analyses revealed skeletal deformities of 13del mice were alleviated at p20 stage by ISRIB treatment (n = 3), including length of tibia, femur and spine (spine here indicated by the length of 7 continuous

*Figure 8 continued on next page*

*Figure 8 continued*

vertebrae consisting of the last sacral and six tail vertebrae), pelvic bone deformation ($\theta 1$: the angle between ilium and pubis), *Coxa Vara* ($\theta 2$: the angle between the proximal head and the shaft of the femur) and *Genu Varum* ($\theta 3$: the angle between proximal head and distal head of tibia). The skeletal phenotypes in vehicle- and ISRIB-treated 13del groups were compared, and the significance of changes was determined by one-way ANOVA test. *: p-value<0.05, **: p-value<0.005, ***: p-value<0.0005. (D–E) Rescue of growth plate abnormalities in 13del mice by the treatment of ISRIB at the p10 and p20 stages, shown by histology (a–a") and in vivo expression profiles of SOX9 (b–b" and c–c"), *Col2a1* (d–d") and *Ppr* (e–e''). Higher magnification of the boxed region was shown to demonstrate the differential expression of SOX9 and the positive cells were indicated by arrows. (Scale Bar = 200 μm). (F) Measurement of the HZ lengths of tested animals at p10 stage (n = 5) and p20 stage (n = 3). The vertical length of the central part of the HZ from 10 different sections of each mouse was averaged. (G) Quantification of SOX9$^+$, *Col2a1* $^+$ and *Ppr* $^+$ cells in HZ of tested animals at p10 stage (n = 5) and p20 stage (n = 3). For each mouse, the number of positive cells was counted and average on five non-adjacent sections. Error bars were shown as S.D. and significance was determined by Two-tailed Mann-Whitney *U*-test. *: p-value<0.05, **: p-value<0.005, ***: p-value<0.0005.

DOI: https://doi.org/10.7554/eLife.37673.018

The following figure supplements are available for figure 8:

**Figure supplement 1.** ISRIB ameliorates skeletal deformities in 13del mice but does not affect the growth of WT mice.

DOI: https://doi.org/10.7554/eLife.37673.019

**Figure supplement 2.** ISRIB ameliorates axial skeleton deformities in 13del mice.

DOI: https://doi.org/10.7554/eLife.37673.020

**Figure supplement 3.** Postnatal administration of ISRIB ameliorates 13del skeletal deformities.

DOI: https://doi.org/10.7554/eLife.37673.021

was barely affected and confirmed the selectivity of ISRIB action (*Figure 9C*). Thus, the effects of ISRIB on ameliorating 13del MCDS phenotype is Xbp1$^S$-independent, and it further supports the redundancy of IRE1/Xbp1 signaling in the pathogenesis of MCDS (*Cameron et al., 2015b*). Also, the transcriptional and translational expression levels of ER chaperone BiP(encoded by *Hspa5*) were relatively attenuated in ISRIB-treated 13del HCs (*Figure 9B and C*), consistent with previous studies showing that PERK-p-eIF2α signaling is required for the activation of BiP upon ER stress (*Harding et al., 2000*; *Scheuner et al., 2001*). Immunostaining with a 13DEL-specific antibody revealed the 13DEL mutant protein were intracellularly accumulated in ISRIB-treated 13del HCs (*Figure 9—figure supplement 1C*), indicating the production and/or degradation of the mutant protein was not affected. Interestingly, inhibition of p-eIF2α signaling by ISRIB at indicated dosage did not induce apoptosis (*Figure 9—figure supplement 1D*) in 13del HCs, with a lowered expression level of ATF4, CHOP and FGF21. This finding highlights the importance of level titration of these factors in the prevention of de-differentiation without causing the death of the stressed chondrocytes. Thus, the effects of ISRIB on ameliorating 13del MCDS phenotype depends explicitly on downregulating p-eIF2α modulated ATF4 induction and its downstream target SOX9.

## Discussion

In this study, we have exploited an *in vivo* model of a congenital developmental disorder and provided mechanistic insight into the question of how the ISR impacts on cell differentiation and fate and importantly, also addressed the possibility of preventive therapy. Our study on a MCDS mouse model demonstrates a direct link between the ISR component of the UPR, PERK-p-eIf2α signaling, and its dominant role in causing reprogrammed chondrocyte differentiation and in mediating their survival. Our study highlights the causative role, *in vivo*, of p-eIF2a induced ATF4 overexpression in HCs, in controlling cell differentiation and survival (*Figure 10*). Given the key roles of SOX9 as an essential modulator in regulating chondrocyte differentiation, proliferation and entry into hypertrophy (*Leung et al., 2011*; *Akiyama et al., 2002*; *Liu et al., 2017*; *Dy et al., 2012*), ATF4-directed transactivation of *Sox9* in HCs where it is not normally expressed, implicates its ectopic expression as the cause of HC reversion to a less differentiated state. Consistent with this possibility is the amelioration of aberrant chondrocyte differentiation when *Sox9* is conditionally ablated in 13del and *C10-Atf4* mice. Ectopic expression of SOX9 can reprogram fibroblasts to chondrogenic fate (*Hiramatsu et al., 2011*), and also acts in a dosage-dependent manner with effects of ectopic overexpression in HCs, ranging from mild expansion of the HZ (*Leung et al., 2011*) to increased apoptosis, defects in vascular invasion and reduced trabecular bone (*Hattori et al., 2010*). Therefore, the

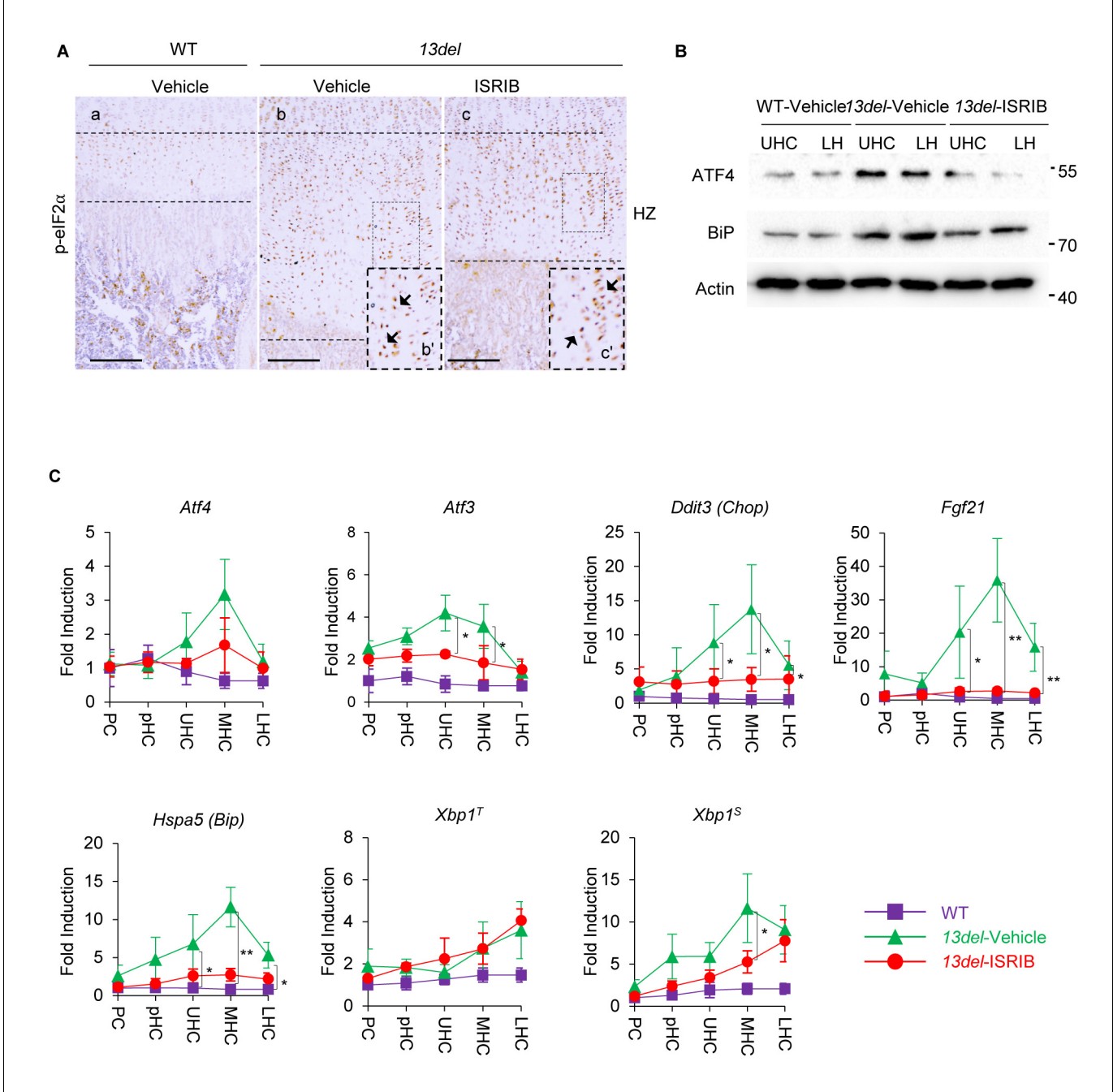

**Figure 9.** The PERK-p-eIF2α signaling pathway is specifically down-regulated by the treatment of ISRIB in 13del HCs. At p10 stage, the expression of p-eIF2α in ISRIB-treated 13del HC was not changed. Higher magnification of the boxed region was shown to demonstrate the expression of p-eIF2 clearly, and the positive staining cells were indicated by arrows. (Scale Bar = 200 μm) (**B**) The vehicle-treated WT, vehicle- and ISRIB-treated 13del p10 HZ were equally fractionated into two chondrocyte populations (UHC and LHC), and protein from each population was isolated. The protein expression level of ATF4 and ER stress sensor BiP were examined via western blot. Beta-actin was used as internal control. (**C**) The WT, vehicle- and ISRIB-treated 13del p10 growth plates were fractionated into different chondrocyte populations (PC, pHC, UHC, MHC and LHC), and total RNA from each population was isolated. The expression levels of *Atf4,* its downstream targets (*Atf3, Ddit3,* and *Fgf21*), ER stress sensor *Hspa5*, total $Xbp1^T$ and its active form $Xbp1^S$ were quantified via qRT-PCR, results were presented as fold induction compared with WT proliferating chondrocytes (PC) from five independent animals of each group. Error bars shown as S.D. and significance between vehicle- and ISRIB-treated HCs was determined Two-tailed Mann-Whitney *U*-test. *: p-value<0.05, **: p-value<0.005.

DOI: https://doi.org/10.7554/eLife.37673.022

The following figure supplement is available for figure 9:

*Figure 9 continued*

**Figure supplement 1.** The p-eIF2α signaling pathway is down-regulated by the treatment of ISRIB.

DOI: https://doi.org/10.7554/eLife.37673.023

impact of the ISR on chondrocyte fate likely depends on the intensity of the ISR and degree of over-expression of SOX9 and chondrocyte differentiation stage.

It is interesting that we found that the Xbp1 component of the IRE1 arm of the UPR did not have a major role in causing reverted chondrocyte differentiation, suggesting differences in utilization of the UPR sensors and that choice of biochemical pathway in cellular response to ER stress may be context dependent. Such a possibility is consistent with another report where a osteogenesis imperfecta COL1A2 G610C mutation causes procollagen I misfolding in the ER of osteoblasts and abnormal osteoblast differentiation, but did not induce the classical UPR mediators BiP or *Xbp1*-spliced form (*Mirigian et al., 2016*). Instead, an unconventional cell stress response was invoked, involving the increase in eIF2α phosphorylation and upregulation of CHOP, key components of the ISR. Although not demonstrated, that report suggests a mechanism for the abnormal osteoblast differentiation in osteogenesis imperfecta, given the link between eIF2α phosphorylation and preferential translation of transcripts encoding factors such as ATF4, a key regulator of osteoblast differentiation (reviewed in [*Greenblatt et al., 2013*]), and CHOP (*Shirakawa et al., 2006*), the ISR emerges as a likely effector of the abnormal osteoblast differentiation in osteogenesis imperfecta.

Human skeletal dysplasia lead to physical disabilities and generates difficulties in education, employment and social life (*Siegel et al., 1991*; *Thompson et al., 2008*). Current treatment options in skeletal disorders are insufficient and may involve controversial surgical procedures such as limb lengthening (*Lie and Chow, 2009*). Growth hormone therapy has been used to treat dwarfism but is clarified to be ineffective for height gain in most congenital skeletal dysplasia, and in some cases with severe spinal deformities, it even results in worsened kyphosis and lordosis (*Kanazawa et al., 2003*), emphasizing the need for a deep understanding of the molecular pathogenesis for effective treatment development. One successful example is the pathogenic elucidation of FGFR3 signaling in achondroplasia (ACH) that led to an effective therapeutic regimen via antagonizing FGFR3 downstream signaling. Based on that, the C-type natriuretic peptide (CNP) analog (BMN111) treatment resulted in a substantial improvement in skeletal parameters in Fgfr3$^{Y367C/+}$mice mimicking ACH and a clinical trial is underway (*Yasoda et al., 2009*). In another case of PASCH, unremitting ER stress was demonstrated to be associated with a self-perpetuating pathological loop between ER stress, inflammation and oxidative stress process. Suppressing inflammation or oxidative stress by aspirin or resveratrol was shown to be effective in rescuing partially limb growth in a COMP mouse model of PSACH (*Posey et al., 2015*).

Therefore, a more profound molecular pathogenic insight into the role of ER stress/ISR in MCDS might provide the basis of developing potential interventions to alleviate the skeletal deformities in this disorder. Early attempts using chemical chaperone sodium butyrate(*Nundlall et al., 2010*) or ER-stress reducing reagent lithium, valproate, or phenylbutyric acid (PBA) (*Posey et al., 2014*) failed to rescue chondrodysplasia in mouse models. In a recent study, the pathological phenotype of another MCDS mouse model harboring *Col10a1* misfolded mutation was partially ameliorated by administration of carbamazepine (CBZ), a drug that induces mutant type X collagen breakdown (*Mullan et al., 2017*). Notably, CBZ treatment of MCDS mice reduced the stress level and attenuated the expression level of ATF4 in HCs (*Mullan et al., 2017*), consistent with our finding that the disease severity is positively correlated with the HC expression level of ATF4. However, how ER stress induced ATF4 expression causes the chondrocyte differentiation defect was not delineated.

Knowledge of the precise signaling pathway effector(s) that is critical for normal skeletal physiology and homeostasis, and mechanisms of causation for the aberrant cell differentiation associated with MCDS and other skeletal disorders is essential for the development of a targeted treatment strategy. Pharmacological inhibition of PERK or PERK-mediated phosphorylation of downstream targets has been reported to be neuroprotective (*Halliday et al., 2015*), addressing the postnatal impact of activating the PERK signaling in neurodegeneration and also highlights the importance of determining the precise mechanism of causality of this pathway under specific scenarios of degeneration versus cell differentiation in development (*Moreno et al., 2013*). Interestingly, a recent study

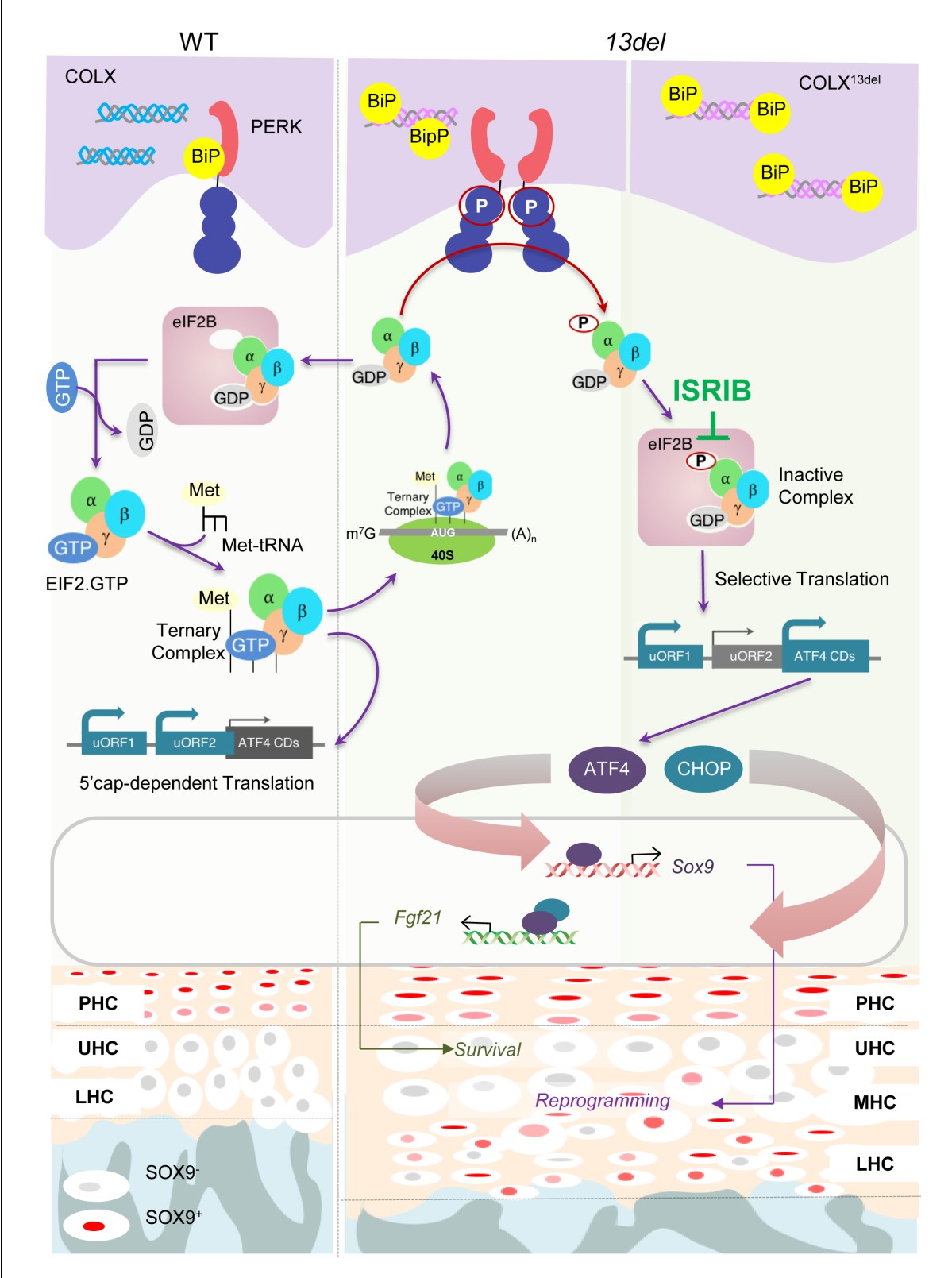

**Figure 10.** A schematic model for PERK signaling pathway in modulating skeletal phenotype in MCDS 13del mice. ISRIB eased the MCDS symptoms by inhibiting the ectopic expression of ATF4, induced by inactive eIF2(p)-eIF2B complex, and the consequent expression of *Sox9*.

DOI: https://doi.org/10.7554/eLife.37673.024

identified a novel mechanism by which the GCN2-regulated ISR-p-elf2α antagonizes Bone Morphogenetic Protein (BMP) signaling, although direct effects on translation and indirectly via *crc* (dATF4) during development in *Drosophila* (*Malzer et al., 2018*). Given the vital role of BMP in regulating multiple developmental processes in organisms from insects to mammals, including chondrocyte/osteoblast differentiation, our study adds to the growing evidence of the importance of the ISR not only in disease scenarios but in regulating normal differentiation and skeletal homeostasis. Taken together, it was therefore essential to address whether manipulating the PERK-p-elf2α pathway in the scenario of a congenital abnormality caused by the deleterious impact of the ER stress/ISR on skeletal development, could be useful.

Our findings also illustrate the importance of where in the PERK pathway a therapeutic approach should be taken. Global inhibition of CHOP will not be effective because its inactivation exacerbates the defect. By the same token, given the importance of ATF4 to normal development simply preventing its expression globally would not work therapeutically. Instead, a viable approach would be to target the translational control that leads to its overexpression.

Here, we demonstrate that for a disorder that is caused by the impact of the p-eIF2α signaling on cell fate and differentiation during development, a practical therapeutic approach is to target early in pathway, at the level of p-eIF2α-mediated translation control by specific inhibitors, such as ISRIB (*Figure 10*). By that, we have shown that ISRIB enfeebled the induction of ATF4 and its downstream target SOX9, prevented the aberrant cell differentiation and consequential dwarfism without obvious side effects. Previously, the selectivity of ISIRIB in targeting translational control of p-eIF2α has been studied by different reports, where it has been used in studies in mice such as treatment of neurodegeneration (*Halliday et al., 2015*; *Wong et al., 2018*), aggressive prostate cancer (*Nguyen et al., 2018*), in enhancing memory and memory consolidation in mice (*Sidrauski et al., 2013*), cognitive deficits after traumatic brain (*Chou et al., 2017*) and in rescuing impaired sociability and anxiety-like behavior (*Kabir et al., 2017*). ISRIB has also been used to address expression of OPHN1 as a translational downstream target of p-eIF2α that is associated with mGluR-LTD-linked cognitive disorders (*Di Prisco et al., 2014*). Here, the effect of ISRIB on the aberrant differentiation of ER-stressed HCs reveals its action on antagonizing preferential translation of ATF4, and consequent SOX9 induction to reverse or prevent the abnormal cell fate change in disease.

Interestingly, by the treatment of ISRIB at the indicated dosage, the enfeebled action of p-eIF2α did not induce apoptosis, with lowered but detectable expression of ATF4, CHOP and downstream target FGF21. This finding reveals, in the context of the 13del mutation, the dominance of dedifferentiation in the pathogenesis of MCDS. Notably, PERK was reported to have a role in pancreatic β cells under physiological ER stress conditions, via regulating β cell proliferation/development, proinsulin production/trafficking, insulin secretion and protecting the cells from tolerable ER stress in an ATF4-dependent manner. However, continuous eIF2α phosphorylation induces β cell death via activating ATF4-CHOP pathway (*Oslowski C and Urano, 20102011*). These findings strongly suggest PERK/ISR signalling pathway may behave as a binary switch between life and death in organ homeostasis. By finding the dosage of ISRIB that titrates ATF4-CHOP levels and is protective for de-differentiation, without causing the death of the stressed chondrocytes, the dualism inherent in the PERK signaling may be exploited therapeutically. Our study may also reflect the possibility that other potential translational downstream target(s) of p-eIF2α may be involved in the cell death induction in 13del HCs upon stress and act upstream of CHOP and FGF21.

The inability of FGF21 expressing 13del HCs to protect 13del;*Fgf21*[-/-] HCs from apoptosis in mouse chimeras suggests FGF21 cell autonomously protects HCs from apoptosis. Interestingly, our microarray analyses show that *blk*, the gene encoding beta-klotho, which is required for FGF21-mediated receptor binding and activation (*Kurosu et al., 2007*), is not expressed in 13del HCs, suggesting that ISR-induced FGF21 may function in a FGFR1/beta-klotho-independent manner to enable the HCs survival. The lack of *blk* expression in 13del HCs suggest the pro-survival role of FGF21 in ER stressed HCs could be non-canonical. It has been reported that FGF21 reduces the concentrations of the active form of STAT5, a major mediator of growth hormone actions, lowers the expression of IGF-1, but induces the expression of IGF-1 binding protein one to blunt growth hormone signaling (*Inagaki et al., 2008*). Our microarray data show that the expression levels of these FGF21 targets were comparable between WT and 13del HCs. FGF21 is a biomarker for mitochondrial translation and mtDNA maintenance disorders, and a stress-induced myokine in mitochondrial diseases (*Lehtonen et al., 2016*). FGF21 is also proposed to be a critical metabolic

mediator of ISR presumably causing systemic metabolic improvements (*Keipert et al., 2014*). Moreover, FGF21 has been reported to protect the mouse liver and brain against d-gal-induced oxidative stress and apoptosis via activating Nrf2 and PI3K/Akt pathways (*Yu et al., 2015a2015a*; *Yu et al., 2015b2015b*). Analyses of the proteome of 13del HCs has implicated the potential role of the ER-mitochondria pathway in chondrocyte survival by a mechanism whereby changes in ER-mitochondria communication reduce import of calcium coupled to maintenance of mitochondrial membrane polarity (*Kudelko et al., 2016*). Our transcriptome data show that oxidative stress was triggered in 13del HCs (*Figure 1D and E*). It will therefore be important to study the intrinsic roles of ISR-induced FGF21 in mitochondria homeostasis and in regulating oxidative stress-related apoptosis.

It is also noteworthy that while ISRIB treatment significantly improved the pathological phenotype of 13del MCDS mice, mutant type X collagen was still detectable and therefore some residual ER stress was present. Consistent with residual ER stress in ISRIB-treated HCs is the persistent expression of ATF4, albeit at lowered levels, which may account for incomplete rescue. Likewise rescue by CBZ treatment was only partial (*Mullan et al., 2017*). It would be important in future to test whether a strategy of addressing both aberrant cell differentiation and enhancing mutant protein degradation by combined treatment with ISRIB and CBZ, could achieve complete rescue of MCDS and other skeletal disorders associated with the UPR/ISR. In addition, the relative poor solubility of ISRIB might have limited its bioavailability *in vivo*. A recent study of ISRIB treatment in prostate cancer model showed improved bioavailability of ISRIB by using HPMT vehicle solution (0.5% w/v hydroxypropyl-methylcellulose dissolved in water plus 0.2% v/v Tween80, adjusted to pH 4) (*Nguyen et al., 2018*). For future studies, it would be important to test whether ISRIB dissolved in HPMT will result in improved alleviation of MCDS and other ISR-related disorders.

Our studies have shown that ISRIB is effective in ameliorating the chondrodysplastic defects by treatment in the fetal and postnatal periods where the growth plate is most active as rapid growth occurs. We did not test if ISRIB treatment would be able to reverse the dwarfism, if given after maturity and cessation of skeletal growth. It should be noted, however, that the expression of 13del transgene reduces postnatally around 3 weeks and stops expressing in HCs at around 4 weeks (*Tsang et al., 2007*). We previously showed that postnatally, the intensity of ER stress signalling correlates with levels of 13del expression (*Tsang et al., 2007*). This would mean that in a scenario where the ER stress is relieved, treating adult 13del mice with ISRIB may not be effective in overcoming the dwarfism. Therefore, prevention of aberrant HC differentiation during the growth phase should be more effective therapeutically than once growth has ceased in maturity.

Our study has implications for other disorders. ER stress and the ISR are also implicated in intervertebral disc degeneration, which is very common in humans, and a major cause of low back pain (*Xu et al., 2017*; *Zhao et al., 2010*). Therefore, an important question arising is whether activation of the ISR and its associated ectopic expression of ATF4 and its downstream targets such as *Sox9* may underlie common skeletal diseases such as intervertebral disc degeneration and osteoarthritis.

The direct activation of *Sox9* by ATF4, also may have broad implications for other diseases. SOX9 is a potent transcription factor with critical roles in cell fate determination, not only in chondrocytes but also in many different cell types, notably stem cells (dermal papilla, gonads, intestinal and neural, etc.)(*Posey et al., 2014*; *Pritchett et al., 2011*) and is activated in other common and acquired diseases such as cancer and fibrosis (*Pritchett et al., 2011*; *Athwal et al., 2017*). Ectopic expression of SOX9 has been widely reported to mediate ECM deposition, such as collagens and other structural proteins, in the pathology of fibrosis in multiple organs, including the liver (*Athwal et al., 2017*), kidney (*Kang et al., 2016*; *Li et al., 1864*), lung and myocardium (*Lacraz et al., 2017*). These observations of ectopic and/or over expression of SOX9 raise questions about the upstream role of the ISR in the pathogenesis and/or progression of these diseases. Also, SOX9 directly activates the ER stress transducer BBF2H7 (*Hino et al., 2014*) with added implications for those diseases where ectopic or over-expression of SOX9 has been described (*Pritchett et al., 2011*; *Athwal et al., 2017*; *Lacraz et al., 2017*) Therefore, whether targeting eIF2a-mediated translation control could be a general therapeutic strategy for ISR-associated common human diseases is another important question that should be addressed in future.

# Materials and methods

## Key resources table

| Reagent type (species) or resource | Designation | Source or reference | Identifiers | Additional information |
|---|---|---|---|---|
| Strain, strain background (*Mus musculus*) | *13del* | PMID: 17298185 | | Maintained in F1 (C57BL/6 x CBA) background |
| Strain, strain background (*Mus musculus*) | *C10-Atf4* | This paper | | |
| Strain, strain background (*Mus musculus*) | *Ddit3$^{-/-}$* | PMID: 9531536 | | Gift from Prof. David Ron's lab (University of Cambridge, UK) |
| Strain, strain background (*Mus musculus*) | *Fgf21$^{-/-}$* | PMID:19589869 | | Gift from Prof. Nobuyuki Itoh's lab (University of Kyoto, Japan) |
| Strain, strain background (*Mus musculus*) | *Col10a1$^{Egfp}$* | PMID:25092332 | | |
| Strain, strain background (*Mus musculus*) | *Sox9$^{flox/flox}$* | PMID:12414734 | | Gift from Prof. Andreas Schedl's lab (Institute of Biology Valrose, France) |
| Strain, strain background (*Mus musculus*) | *Col10a1$^{Cre}$* | PMID:25092332 | | |
| Cell line (*Mus musculus*) | ATDC5 | RCB0565; PMID:8609176 | Chisa Shukunami | |
| Cell line (*Mus musculus*) | MEF | Isolated from E13.5 mouse embryo (F1) | | Mouse Embryonic Fibroblast |
| Cell line (*Mus musculus*) | NIH 3T3 | ATCC CRL-1658 | | Mouse fibroblast suitable for transfection |
| Transfected construct | pSox9-2.7K | This paper | | 2.7K Sox9 promoter inserted into pGL3 basic vector (Promega) |
| Transfected construct | pSox9-1.8K | This paper | | 1.8K Sox9 promoter inserted into pGL3 basic vector (Promega) |
| Transfected construct | pSox9-0.8K | This paper | | 0.8K Sox9 promoter inserted into pGL3 basic vector (Promega) |
| Transfected construct | pSox9-1.8M1 | This paper | | 1.8K Sox9 promoter with one mutated ATF4 binding site inserted into pGL3 basic vector (Promega) |
| Transfected construct | pSox9-1.8M2 | This paper | | 1.8K Sox9 promoter with one mutated ATF4 binding site inserted into pGL3 basic vector (Promega) |
| Transfected construct | pSox9-1.8M3 | This paper | | 1.8K Sox9 promoter with two mutated ATF4 binding sites inserted into pGL3 basic vector (Promega) |
| Transfected construct | pFgf21-Luc1 | This paper | | 2K Fgf21 promoter inserted into pGL3 basic vector (Promega) |
| Transfected construct | pFgf21-Luc2 | This paper | | 1.2K Fgf21 promoter inserted into pGL3 basic vector (Promega) |
| Transfected construct | pFgf21-Luc3 | This paper | | 0.8K Fgf21 promoter inserted into pGL3 basic vector (Promega) |
| Transfected construct | pFgf21-Luc4 | This paper | | 0.4K Fgf21 promoter inserted into pGL3 basic vector (Promega) |

*Continued on next page*

*Continued*

| Reagent type (species) or resource | Designation | Source or reference | Identifiers | Additional information |
|---|---|---|---|---|
| Transfected construct | pFgf21-M1 | This paper | | 1.2K Fgf21 promoter with one mutated ATF4 binding site inserted into pGL3 basic vector (Promega) |
| Transfected construct | pFgf21-M2 | This paper | | 1.2K Fgf21 promoter with one mutated ATF4 binding site inserted into pGL3 basic vector (Promega) |
| Transfected construct | pFgf21-M3 | This paper | | 1.2K Fgf21 promoter with two mutated ATF4 binding site inserted into pGL3 basic vector (Promega) |
| Antibody | anti-ATF4 | sc-200 | Santa Cruz | rabbit IgG |
| Antibody | anti-CHOP | sc-575 | Santa Cruz | rabbit IgG |
| Antibody | anti-SOX9 | AB5535 | Millipore | rabbit IgG |
| Antibody | anti-ATF3 | HPA001562 | Sigma | rabbit IgG |
| Antibody | anti-FGF21 | 42189 | AIS | rabbit IgG |
| Antibody | anti-PPP1R15A | sc-825 | Santa Cruz | rabbit IgG |
| Antibody | anti-BiP | ADI-SPA-826 | ENZO | rabbit IgG |
| Antibody | anti-beta Actin | A2228 | Sigma | mouse IgG |
| Commercial assay or kit | In Situ Cell Death Detection Kit | 12156792910 | Roche | TUNEL assay |
| Commercial assay or kit | Dual-Luciferase Reporter Assay System | E1910 | Promega | Luciferase assay |
| Commercial assay or kit | BrdU staining kit | 93–3943 | Thermo Fisher Scientific | |
| Chemical compound, drug | ISRIB | SML0843 | Sigma | |
| Software, algorithm | RMA algorithm | R Bioconductor | | Robust Multi-chip Average |
| Software, algorithm | k-Means Clustering algorithm | DOI: https://doi.org/10.1016/j.patrec.2009.09.011 | | |
| Software, algorithm | Gene Ontology database | PMID:10802651 | | |
| Software, algorithm | HOMER software package | PMID:20513432 | | |
| Software, algorithm | Bowtie program | PMID:19261174 | | |
| Software, algorithm | Picard toolkit of Broad Institute | https://tldrlegal.com/license/mit-license | MIT | |
| Software, algorithm | Prism | | GraphPad software | |

## Genetically modified mice and mutant analysis

The 13del transgenic mice have been described previously (*Tsang et al., 2007*) and were maintained in F1 (C57BL/6 x CBA) background. The *Ddit3 (Chop)*-null mice (gift of David Ron) and Fgf21-null mice were reported previously (*Zinszner et al., 1998*; *Hotta et al., 2009*). The Sox9-flox mice were a gift from Andreas Schedl (*Akiyama et al., 2002*). A BAC transgene vector (*Col10a1-Atf4*-IRES-*Egfp*, abbreviated as *C10-Atf4*) was generated using the BAC clone RPCI23-194I3 (113), which carries the *Col10a1* gene, and by inserting the coding region of *Atf4* at the *Col10a1* ATG codon in exon 2. Included in the transgene are 154 kb and 35 kb of 5′ and 3′ sequences, respectively, relative to *Col10a1* exons. This BAC vector can recapitulate the expression pattern of *Col10a1* (*Yang et al., 2014*). *C10-Atf4* transgenic mice were generated by pronuclear injection into F1 C57/Bl6 x CBA zygotes and maintained in the same F1 background. The primers used for *C10-Atf4* mouse

genotyping were listed in *Supplementary file 7*. Animal care and experiments performed were in accordance with the protocols approved by the Committee on the Use of Live Animals in Teaching and Research of the University of Hong Kong.

Comparisons of *13del* vs. *13del;Ddit3⁻/⁻*, *13del* vs. *13del;Sox9^{c/c}*, *C10-Atf4* vs. *C10-Atf4;Sox9^{c/c}* and *13del* vs. *13del; Fgf21⁻/⁻* were conducted between littermates only, in which any animal of desired genotype were only compared to its control littermates. For each compound mutant line, double or triple mice were intercrossed for >10 generations before any evaluation to avoid issues of differences in genetic background. The gender of double or triple mutant test animals was randomized when sampling for analysis.

The *13del*, *Col10a1^{cre}*, *Sox9^{flox/flox}* and *Fgf21⁻/⁻* were genotyped according to previous studies (*Tsang et al., 2007*; *Yang et al., 2014*; *Akiyama et al., 2002*; *Hotta et al., 2009*), respectively.

## RNA preparation and microarray analysis

The fractionation methodology was established as described previously (*Tan et al., 2018*). In brief, chondrocyte sub-populations in transverse sections of the proximal tibia of 10-day-old WT or 13del mouse were obtained by cryosectioning. 5-micron sections were prepared and pooled into fractions consisting of 10 sections per fraction to ensure separation of each cell type in the growth plate. Samples were dissolved in Trizol reagent (Invitrogen) for RNA extraction. To guide the sub-division of chondrocyte populations into zones, every 10th section was examined histologically and 10% of the RNA isolated from sections selected at regular intervals was used for the detection of known growth plate markers by RT-PCR analyses (*Figure 1—figure supplement 1B*).

Total RNA was extracted and hybridized to Mouse Genome 430 2.0 Gene Chip (Affymetrix). Gene expression data for each sample in triplicate were normalized using Robust Multi-chip Average (RMA) algorithm in *R* Bioconductor package. The *k*-Means Clustering algorithm(*Quackenbush, 2001*; *Jain, 2010*) was used to identify the distinct expression patterns of genes in WT and 13del growth plates, which aims to partition *n* observations into *k* clusters in which each observation belongs to the cluster with the nearest mean, serving as a prototype of the cluster. The Gene Ontology analysis was performed for each cluster of genes by using the Gene Ontology database (*Ashburner et al., 2000*) and the David Web Tools (*Dennis et al., 2003*).

## HOMER motif discovery

The DNA-binding motif enrichment analysis was performed by using HOMER software package (*Heinz et al., 2010*). The DNA sequences flanking the genes' transcription start sites 2 kb up- and downstream were extracted from the mouse reference genome assembly (mm9). The HOMER, the TRANSFAC (*Wingender et al., 1996*) and the ISMARA (*Balwierz et al., 2014*) transcription factor databases were integrated to create the TF binding motif library for screening. The DNA sequences of the interrogated gene sets were compared with those extracted from the remainder gene sets to identify the differentially enriched DNA binding motifs and the TFs.

## ChIP-sequencing data analysis

The ATF4 and CHOP ChIP-sequencing datasets (*Han et al., 2013*) were downloaded from the GEO database (GSE35681). The DNA sequences were aligned to the mm9 mouse reference genome assembly with Bowtie program (*Langmead et al., 2009*). The analysis of coverage signal intensity and peak detection were performed by using Picard toolkit of Broad Institute (MIT) (https://tldrlegal.com/license/mit-license). The binding peaks located within 10 kb up or downstream of the TSS in each target gene were identified for statistical analysis in each cluster.

## Histological and immunofluorescence analyses

Methods used were as described previously (*Tsang et al., 2007*). In brief, limbs were fixed in 4% PFA, followed by demineralization in 0.5M EDTA (pH 8.0) before embedding in paraffin. Slides were stained with Alcian Blue for cartilage matrix and Fast Red for nuclei. Immunofluorescence was performed using antibodies against ATF4 (sc-200, Santa Cruz), ATF3 (HPA001562, Sigma), CHOP (sc-575, Santa Cruz), PPP1R15A (sc-825, Santa Cruz), FGF21 (42189, AIS) and Sox9 (AB5535, Millipore).

## FAST staining

FAST staining referred to a multidye staining procedure using fast green, Alcian blue, Safranin-O, and tartrazine and was performed as described previously(*Leung et al., 2009*).

## In-situ hybridization

In-situ hybridization was performed as previously described(*Wai et al., 1998*), using [$^{35}$S]UTP-labeled ribopobes for *Col10a1*, *Col2a1*, *Hspa5 (Bip)*, *13del*(*Tsang et al., 2007*), *Ihh* (from A. McMahon), *Sox9*(*Ng et al., 1997*) and the *PTHrP* receptor (*Ppr*) (from H. Kronenberg). The probes for *Atf4*, *Atf3*, *Ddit3 (Chop)*, *Ero1l* and *Fgf21* were mouse cDNA fragments, generated by RT-PCR from growth plate total RNA. The primers used for probe synthesis were listed in *Supplementary file 7*.

## Generation of *13del;Col10a1$^{Egfp}$Egfp13 del;Fgf21$^{-/-}$* chimeras

We utilized compound mutants carrying the 13del transgene and a *Col10a1$^{Egfp}$* allele [*Egfp* knocked into the *Col10a1* gene (*Yang et al., 2014*) so that all 13del HCs are marked by EGFP expression. We created mouse chimeras by aggregating morulae from *13del;Col10a1$^{Egfp}$* and *13del;Fgf21$^{-/-}$*mice. The chimeras (agouti/black) generated were genotyped for the different alleles by PCR. The HCs from *13del;Col10a1$^{Egfp}$* with FGF21 expression were visualized by EGFP or by immunostaining for EGFP on cryosectioned growth plates.

## TUNEL assay and BrdU incorporation

Methods used are as described in *Tsang et al. (2007)*. In brief, apoptotic cells in the growth plate of examined animals were detected by in situ terminal deoxynucleotidyltransferase deoxyuridine triphosphate nick end labeling (TUNEL) assay using the In Situ Cell Death Detection Kit (Roche) following the manufacturer's instructions. Cell proliferation activity was analyzed using BrdU labeling assay. Mice were injected intraperitoneally with 200 µg of BrdU per gram of body weight two hours before sacrifice. Following fixation, BrdU in paraffin sections was detected using a BrdU Staining Kit (93–3943, Thermo Fisher).

## Chromatin immunoprecipitation (ChIP) assay

The protocol used for ChIP was adapted from the instructions of ChIP Assay Kit (Millipore). Cultured cells or limbs dissected from E15.5 WT and *C10-Atf4* embryos were homogenized and crosslinked. DNA was sonicated and immunoprecipitated with rabbit anti-ATF4 (sc-200, Santa Cruz Biotechnology) or rabbit anti-CHOP (sc-575, Santa Cruz Biotechnology) antibody. The pull-down DNA was purified and analyzed by PCR.

## Protein extraction and immunoblot analysis

Cartilages isolated from the mice were pulverized in liquid nitrogen and then lysed with RIPA buffer. The lysate was subjected to SDS-PAGE under reducing conditions and probed with FGF21 and beta-actin antibody. For drug-treated animals, the proximal part of the tibia was embedded in RIPA buffer for cryosection. Transverse sections (5 µm thick) were cut and pooled into fractions consisting of 10 sections per fraction to ensure separation of each cell type in the growth plates before further lysed in RIPA. The lysate was subjected to SDS-PAGE under reducing conditions and probed with ATF4, BiP (ADI-SPA-826, Enzo) and beta-actin (A2228, Sigma) antibody.

## Dual-luciferase reporter assay

Luciferase assays were conducted using a dual luciferase reporter assay kit (Promega), according to the manufacturer's protocol. Different promoter fragments of *Sox9* or *Fgf21* were cloned into a pGL3-basic vector (Promega) to drive the expression of firefly luciferase. ATDC5 (RCB0565, RIKEN, Wako, Japan) or NIH3T3 cells (ATCC CRL-1658) were plated at $2 \times 10^4$ cells/well in 24-well plates. After 18 hr incubation, the cells were transfected with tested constructs with *Renilla* luciferase vector, which served as an internal control. Data presented are ratios of Luc/Renilla activity from at least three different experiments, and each experiment was performed in triplicate for each DNA sample.

## Quantitative PCR

Quantitative PCR was performed using SYBR-Green master mixture according to the manufacturer's instruction (Takara). Appropriate amounts of cDNA (or DNA) and primers were mixed with distilled water up to 10 μl and combined with an equal amount of SYBR-Green master mixture. The reaction was run on the StepOne Real-Time PCR system (Applied Biosystems, A and B). The Ct (cycle threshold) is defined as the cycle number required for the fluorescent signal to cross the threshold. The relative expression levels of target genes are calculated by normalizing to the expression level of GAPDH using delta-delta-Ct (Relative expression level = $2^{-(Ct_{target}- Ct_{Gapdh})}$). The melting curve was also measured to detect the specificity of the primers. The primers used for qRT-PCR were listed in *Supplementary file 7*.

## ISRIB treatments

ISRIB (SML0843, Sigma) was dissolved in DMSO to make a 5 mg/ml stock and stored at 4-degree. Animals were intraperitoneally injected with ISRIB (*Di Prisco et al., 2014*; *Sidrauski et al., 2013*) (2.5 mg/kg, freshly diluted in 0.9% saline) or vehicle (5% DMSO in saline) from E13.5 till p20 stage or from p0 till 4-week stage. The animals were collected at p10 and p20 stages for further analysis.

## Radiography of mouse skeleton

Mice were anesthetized before radiography using digital Faxitron system (UltraFocus) at 20kVA for 5 s exposure.

## Statistical analyses

No statistical methods were used to predetermine sample size. Statistical methods used are detailed in the figure legends. We used two-tailed Mann-Whitney *U*-test to establish statistical significance in all mouse phenotypic analysis, unpaired Two-tailed Student's *t*-test in the luciferase assay, and one-way ANOVA in the ISRIB-treatment studies, p<0.05 was considered statistically significant.

## Data availability

All primary microarray data are deposited into Gene Expression Omnibus (GEO) website (Accession Number GSE99306). Source Data for all figures are provided with the paper and supplementary files.

## Acknowledgements

This work was supported by grants from the Hong Kong Research Grants Council grants AoE/M-04/04, T12-708/12 N, RGC-NSFC 31361163004/N_HKU703/13, and HKU 760411M. We thank David Ron for sharing mouse resources and advice, Reinhard Faessler for critical discussion and Pak Sham for advice on statistical methodology.

## Additional information

### Funding

| Funder | Grant reference number | Author |
|---|---|---|
| Research Grants Council, University Grants Committee | AoE/M-04/04 | Kathryn Song Eng Cheah |
| Research Grants Council, University Grants Committee | T12-708/12-N | Kathryn Song Eng Cheah |
| Research Grants Council, University Grants Committee | N_HKU703/13 | Kathryn Song Eng Cheah |
| National Natural Science Foundation of China | 31361163004 | Kathryn Song Eng Cheah |
| Research Grants Council, University Grants Committee | HKU 760411M | Danny Chan |

The funders had no role in study design, data collection and interpretation, or the decision to submit the work for publication.

## Author contributions

Cheng Wang, Conceptualization, Resources, Data curation, Supervision, Funding acquisition, Investigation, Project administration, Writing—review and editing; Zhijia Tan, Ben Niu, Conceptualization, Data curation, Formal analysis, Validation, Investigation, Visualization, Methodology, Writing—original draft, Writing—review and editing; Kwok Yeung Tsang, Data curation, Software, Formal analysis, Investigation, Methodology, Writing—review and editing; Andrew Tai, Conceptualization, Data curation, Investigation, Visualization, Writing—review and editing; Wilson C W Chan, Data curation, Investigation, Visualization, Writing—review and editing; Rebecca L K Lo, Conceptualization, Data curation, Writing—review and editing; Keith K H Leung, Conceptualization, Data curation, Validation, Methodology, Writing—review and editing; Nelson W F Dung, Data curation, Investigation, Methodology, Writing—review and editing; Nobuyuki Itoh, Investigation, Methodology, Writing—review and editing; Michael Q Zhang, Resources, Data curation, Investigation, Writing—review and editing; Danny Chan, Conceptualization, Data curation, Software, Supervision, Investigation, Visualization, Methodology, Writing—review and editing; Kathryn Song Eng Cheah, Conceptualization, Resources, Data curation, Supervision, Funding acquisition, Investigation, Visualization, Project administration, Writing—review and editing

## Author ORCIDs

Zhijia Tan https://orcid.org/0000-0003-2295-5169
Wilson C W Chan http://orcid.org/0000-0003-2381-2805
Michael Q Zhang http://orcid.org/0000-0002-7408-1830
Danny Chan http://orcid.org/0000-0003-3824-5778
Kathryn Song Eng Cheah http://orcid.org/0000-0003-0802-8799

## Ethics

Animal experimentation: Animal care and experiments performed were in accordance with the protocols approved by the Committee on the Use of Live Animals in Teaching and Research of the University of Hong Kong (CULATR number: 3981-16).

## Decision letter and Author response

Decision letter https://doi.org/10.7554/eLife.37673.038
Author response https://doi.org/10.7554/eLife.37673.039

# Additional files

## Supplementary files

• Supplementary file 1. List of differentially expressed genes in clusters I, II, III and IV. (Related to *Figure 1*)
DOI: https://doi.org/10.7554/eLife.37673.025

• Supplementary file 2. Genes involved in enriched biological processes in cluster I, II, III and IV. (Related to *Figure 1*)
DOI: https://doi.org/10.7554/eLife.37673.026

• Supplementary file 3. Genes involved in enriched pathways in cluster I, II, III and IV. (Related to *Figure 1*)
DOI: https://doi.org/10.7554/eLife.37673.027

• Supplementary file 4. Differentially expressed genes between 13del and published MCDS transcriptome. (Related to *Figure 1*)
DOI: https://doi.org/10.7554/eLife.37673.028

• Supplementary file 5. Enriched motifs identified in cluster I genes using HOMER software package. (Related to *Figure 1*)

DOI: https://doi.org/10.7554/eLife.37673.029

• Supplementary file 6. List of ATF4 and CHOP target genes in each cluster. Genes containing ATF4 and CHOP binding peaks within 10 kb from TSS were listed. (Related to *Figure 1*)
DOI: https://doi.org/10.7554/eLife.37673.030

• Supplementary file 7. Primers used for genotyping, probe synthesis and qRP-PCR. (Related to Materials and methods)
DOI: https://doi.org/10.7554/eLife.37673.031

• Transparent reporting form
DOI: https://doi.org/10.7554/eLife.37673.032

## Data availability

All primary microarray data are deposited into Gene Expression Omnibus (GEO) website (accession no. GSE99306).

The following dataset was generated:

| Author(s) | Year | Dataset title | Dataset URL | Database, license, and accessibility information |
|---|---|---|---|---|
| TanZ, Wang C, Niu B | 2017 | Expression data of chondrocytes subpopulations from WT, 13del and 13del:Chop-/- mice at P10 stage | https://www.ncbi.nlm.nih.gov/geo/query/acc.cgi?acc=GSE99306 | Publicly available at the NCBI Gene Expression Omnibus (accession no: GSE99306) |

The following previously published dataset was used:

| Author(s) | Year | Dataset title | Dataset URL | Database, license, and accessibility information |
|---|---|---|---|---|
| Kaufman RJ, Han J, Sartor MA | 2013 | ChIP-seq and mRNA-seq experiment to find the direct target genes of ATF4 and CHOP | https://www.ncbi.nlm.nih.gov/geo/query/acc.cgi?acc=GSE35681 | Publicly available at the NCBI Gene Expression Omnibus (accession no: GSE35681) |

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
