## [Decision Letter]

Thank you for submitting your article "Inhibiting the Integrated Stress Response Prevents Aberrant Chondrocyte Differentiation, alleviating Chondrodysplasia" for consideration by *eLife*. Your article has been reviewed by four peer reviewers, including Clifford J Rosen as the Reviewing Editor and Reviewer #1, and the evaluation has been overseen by Harry Dietz as the Senior Editor.

The reviewers have discussed the reviews with one another and the Reviewing Editor (BRE) has drafted this decision to help you prepare a revised submission.

Summary:

There was general enthusiasm for this manuscript from all four reviewers. It breaks new ground in understanding the reversion of chondrocyte hypertrophy via the PERK pathways and enhanced secretion of ATF4 and CHOP and transactivation of *Sox9*. However, there were concerns that need to be addressed in the full revision as noted below:

Essential revisions:

1) Clarify whether FGF-21 works in a cell autonomous or non-cell autonomous manner on chondrocytes and whether there is a phenotype in the *Fgf21^-/-^*mice.

2) Redo the statistics using ANOVA rather than t- tests, which are not appropriate.

3) Specify the selectivity of ISRIB and how it was solubilized; also address whether the authors used it in older mice at skeletal maturity.

4) Clarify mouse backgrounds, sex, numbers and age.

Reviewer #1:

In this paper, Wang and colleagues exploited a mouse model of Metaphyseal Chondrodysplasia type Schmid (MCDS) to examine provide the impact of the (Integrated Stress Response) ISR on cell fate. They showed the protein kinase RNA-like ER kinase (PERK) pathway that mediated preferential synthesis of ATF4 and CHOP, dominated the mechanism of dysplasia by reverting chondrocyte differentiation via ATF4 directed transactivation of *Sox9*. Chondrocyte survival is enabled by CHOP, often considered a death promoter, and transactivation of *Fgf21* by the dual action of CHOP and ATF4. Treatment of mutant mice with a chemical inhibitor of PERK signaling prevented the differentiation defects and ameliorated chondrodysplasia. The authors concluded that inhibition of the ISR could become the cornerstone of a rationale therapeutic strategy for stress-induced skeletal disorders.

Overall the paper is well written, the methods are clearly spelled out and the genetic studies are well described and logical. The mechanisms of the integrated stress response are complex, but the authors have systematically approached this by using network and pathways analysis to identify downstream pathways related to PERK, particularly ATF4 and CHOP to induce de-differentiation but excluding XBP1 in the UPR. The mouse model of 13del MCDS provides a useful model of misfolded mutant collagen X inducing an ER stress suggesting its primary role in the pathogenesis of the mutation. Their data that increases in ATF4 and CHOP through FGF21 contribute to the de-differentiation phenotype are relatively compelling; loss of function and gain of function studies provide further support for their central tenet. However, there are a few concerns:

Of major importance there is no sex mentioned for the mice; the mouse numbers are small (n=5) and the mouse backgrounds are quite different. All this may contribute to some phenotypic differences; The Chop^-/-^ mice are either 129 or CD1 mixed backgrounds, crossed with the 13del which are B6xCBA F1 hybrids; and then the crosses with *Fgf21^-/-^* that are on a B6 background. And there are no data on the *C10-Atf4* Tg. So much more explanation of the crosses is required and how the mixed background that result from these crosses could contribute to the growth plate phenotypes.

FGF21 plays a critical role in the mutational phenotype in respect to survival; it would be useful to show the absence of a growth plate phenotype in the FGF21 null mice. Also, for Figure 6D qRT-PCR should be shown rather than the relative expression by micro array.

It’s not clear from the figure if in the 13del mice only *Sox9* is upregulated; particularly relevant would be changes in Gli 2,3 which have promoter sites for binding by ATF4; moreover, were the gene expression changes checked by qRT-PCR or only by micro array; that needs clarification in the legend.

How specific is ISRIB as an inhibitor of the Integrated stress response?

Also, the rescue of 13del wit ISRIB was not complete and as noted there was still mutant collagen X present. Can the authors provide some insight into other potential mechanisms for the incomplete skeletal rescue if indeed the major pathway is through PERK?

The body length phenotype is of interest in the *C10-Atf4* Tg; is it due to increases in vertebrae growth plate; intervetebral discs? Can the authors elaborate on that phenotype.

Reviewer #2:

I am very supportive of this study, but feel some minor revisions would improve the manuscript.

This study addresses the role of eIF2α phosphorylation and its downstream effects in the pathogenesis of chondrodysplasia caused by a 13 bp deletion within the *Col10a1* gene. The authors identify the unfolded protein response to be induced in bone growth plates from mutant mice; ectopic expression of ATF4 is then shown to phenocopy some features of the chondrodysplasia by driving loss of chondorcyte differentiation; and haploinsufficiency for *Sox9* (a target of ATF4) is shown to rescue the bone defects cause by mutant *Col10a1* or over-expression of ATF4. In contrast, loss of Chop (another ATF4 target) is shown to exacerbate the bone growth defect. Induction of FGF21 by ATF4 and CHOP is shown to promote the survival of hypertrophic chondrocytes expressing mutant *Col10a1*. The eIF2B-activating drug ISRIB is then used to rescue much of the bone phenotype of *Col10a1* mutant mice. Finally, evidence is presented suggesting ISRIB reduces signalling downstream of eIF2α phosphorylation.

This is a substantial piece of work comprising many well-conducted experiments. It will be of interest to researchers in the fields of the integrated stress response and of bone development. The beneficial effects of CHOP on chondrocyte survival are especially interesting since they chime with previous studies (see below).

1) Some of the experiments are presented as single examples (Figure 2A and E, Figure 3A and C, Figure 6A-D, Figure 8) albeit sometimes at multiple time points. Are repeats available and, if so, can the reproducibility of these effects be displayed in a quantitative form with statistical analysis? This is especially relevant to Figure 8B where the data quality is less good than elsewhere in this manuscript.

2) Could the authors comment on any solubility issues with ISRIB that may have impacted its bioavailability? This drug is somewhat notorious for its difficulty to dissolve in biologically compatible solutions.

3) The wording "ATF4 was also insignificantly lowered" subsection “The PERK-p-eIF2α signaling pathway is downregulated explicitly by treatment of ISRIB” should probably be changed if the statistical test showed no difference. There is no need for Atf4 mRNA to be affected since the effects of ISRIB would be on ATF4 translation rather than transcription.

4) I am not convinced that Student's t-tests, which are used frequently herein, are always appropriately used. Statistical advice ought to be sought, and at the very least adjustments made for multiple comparisons. Analysis of variance with a subsequent post-hoc test would seem more appropriate to this reviewer. This is especially true for Figure 8 but is relevant in several other experiments where t-tests alone appear to have been employed.

Reviewer #3:

In this study, the authors have further examined the pathogenesis of Metaphyseal Chondrodysplasia type Schmid (MCDS), using the 13del mouse model. The mutant mice carry a 13 bp deletion in *Col10a1* and mimic the human condition genetically and phenotypically, including intracellular accumulation of misfolded Col X that leads to ER stress, growth plate chondrocyte malfunction and skeletal growth retardation. Using a combination of transcriptome, biochemical and transgenic approaches, the authors found that the protein kinase RNA-like ER kinase (PERK) pathway -a component of the integrated stress response (ISR)- was activated in mutant growth plates and was associated with increased expression of ATF4 and CHOP. Aberrant ATF4 production likely caused broad and excessive *Sox9* expression, delaying growth plate chondrocyte maturation and hypertrophy, causing chondrocyte accumulation and growth plate lengthening, and impairing endochondral ossification and skeletal growth. Indeed, conditional ablation of *Sox9* using ColX-cre mice ameliorated the skeletal and growth plate phenotype of the 13del mice. The authors tested a pharmacological ISR inhibitor that blocks the action of phosphorylated eIF2α and found that systemic drug administration ameliorated the growth plate and skeletal phenotype of juvenile 13del mice. The authors conclude that pharmacological inhibition of the ISR may represent a rational therapeutic strategy for stress-related skeletal conditions.

The study is of significant interest since the pathogenesis of MCDS and related pathologies remains poorly understood. The study describes a very large and impressive body of experimental studies and provides several lines of evidence that the aberrant behavior of growth plate chondrocytes in the 13del mice is due to excessive and lingering expression of *Sox9*, a key transcription factor that is normally tightly regulated to permit growth plate chondrocyte maturation and a seamless transition from hypertrophic cartilage to endochondral bone. Providing evidence that a pharmacological agent targeting ISR elicits beneficial effects on skeletal growth in the mutants adds significant translational medicine value and relevance to the study.

Concerns are minor.

The authors should provide details of how the growth plate chondrocytes were fractionated into 3 or 4 subpopulations before their use in the transcriptome analyses. Because the data are used to reach mechanistic implications, it is important to show that each fraction was in fact composed by a distinct subpopulation. This is particularly important for 13del samples; given that the mutant growth plates contained abnormal distribution of *Sox9-, col2*- and *Ihh*-expressing cells, the cells may have been more difficult to separate into distinct subpopulations if they had intermediate cell phenotypes (volume, shape, etc.).

The data on FGF21 are of particular interest. *Fgf21* was found to be the most up-regulated gene in 13del hypertrophic chondrocytes, and *Fgf21* deletion in 13del mice led to increased apoptosis in the hypertrophic zone. The authors suggest that the excess levels of FGF21 protected the 13del mutant chondrocytes from apoptosis. Do the authors have data on how such protection occurred and whether the fate of hypertrophic cells was altered in the compound mutant mice, including possible consequences on primary spongiosa? Based on data presented, it appears that FGF21 has no role in WT growth plates. Could the authors comment on this?

The apparent beneficial effects of the ISR inhibitor on the 13del skeletal phenotype are promising, but appear to be modest. Also, only juvenile mice (up to p20) were examined. Did the authors try different drug regimens or administration routes to improve efficacy? Could beneficial effects be elicited in older mice up to skeletal maturity around 2-3 months of age? If treatment was stopped at P20 and mutant mice were followed over time, what happened to their skeletal phenotype?

Reviewer #4:

This study uses a mouse model of Metaphyseal Chondrodysplasia type Schmid (MCDS) to investigate how the integrated stress response affects chondrocyte cell differentiation. This is a comprehensive and insightful study that begins to address the molecular and cellular mechanisms by which 13del mutation in *Col10a1* causes skeletal dwarfism. The authors identify protein kinase RNA-like ER kinase (PERK) as a molecule that regulates ATF4 and CHOP to regulate chondrocyte differentiation and survival. The authors show that activation of the unfolded protein response leads to expression of ATF4 and activation of *Sox9* expression, which represses chondrocyte differentiation. They also show that ATF4 and CHOP promote chondrocyte survival by activating FGF21 and that FGF21 is necessary for cell survival. Finally, they show that a small molecule inhibitor, ISRIB, reduces the dwarfism phenotype of 13del mice.

When Atf4 is overexpressed, Col10a1 expression is greatly decreased. How is the HC zone actually defined in this model? Why is *C10-Atf4* not itself downregulated?

If chondrocytes have impaired differentiation, does this mean that they retain features of proliferating or prehypertrophic chondrocytes? Does chondrocyte proliferation extend into this undifferentiated and expanded HC zone?

The expression of *Sox9* could be quantified in Figure 5C. Is expression per cell changed or just number of cells?

Figure 6D does not make sense. The text says 40% decrease in *Fgf21* in 13del; CHOP^-/-^ mice; however, the red line is above the green line in the figure?

The potential role of FGF21 in cell survival is interesting; however, given the long range signaling abilities of FGF21, it should be shown whether FGF21 is acting locally on HCs or indirectly through another factor and non HC tissue. Is beta-klotho expression changed in HCs and is *FGFR1* responsible for supporting cell survival in 13del HCs? Given that *Fgf21* was highly increased, could it have effects other that mediating the ISR? Does FGF21 regulate growth hormone signaling through STAT5, IGF-1, or IGF-1 binding protein (see PMC2575072, PMC3406689).

Although probably beyond the scope of this study, it may be interesting to see how the 13del mouse responds to FGF21 treatment at the growth plate. Is there a direct effect on ERK phosphorylation?

Figure 8B Blot is of low quality. The actin bands are not consistent. This could explain the differences seen between UHC and LHC. Quantification of this expression data would help especially for BIP.

Figure 9. It would be helpful to also indicate the suppressive effect of ISRIB on eIF2B and subsequently ATF4.

Figure 9—figure supplement 1D. Label indicates that TUNEL is Red, however the probe is Green. Change for consistency.

[Editors' note: further revisions were requested prior to acceptance, as described below.]

Thank you for resubmitting your work entitled "Inhibiting the Integrated Stress Response Prevents Aberrant Chondrocyte Differentiation, alleviating Chondrodysplasia" for further consideration at *eLife*. Your revised article has been favorably evaluated by Harry Dietz (Senior Editor) and Clifford Rosen (Reviewing Editor).

The manuscript has been improved and the response to reviewers is complete except for the statistical analysis. This is the only remaining issue prior to acceptance and we ask that the authors more fully justify the use of the Mann-U Whitney test, particularly as it relates to multiple comparisons in Figure 8 vs the more standard and accepted one way or in some cases two way ANOVA. We recommend that you consult a statistician for your response. Once this is resubmitted, we can rapidly make a final decision.

---

## [Author Response]

Essential revisions:1) Clarify whether FGF-21 works in a cell autonomous or non-cell autonomous manner on chondrocytes and whether there is a phenotype in the FGF-21-/- mice.

WT hypertrophic chondrocytes (HCs) do not express FGF21. In our paper we have shown that expression of *Fgf21* is activated as a consequence of ER stress in 13del HCs and this has a survival effect. To determine whether the pro-survival role of FGF21 was cell autonomous or non-cell autonomous in 13del hypertrophic chondrocytes (HCs), we tested the survival of 13del HCs carrying an *Fgf21* null mutation, in mouse chimeras. We utilised compound mutants carrying the 13del transgene and a *Col10a1^Egfp^* allele [*Egfp* knocked into the *Col10a1* gene (Tsang et al., 2007)] so that all 13del HCs are marked by GFP expression. We created mouse chimeras by aggregating *13del;Col10a1^Egfp^*and *13del;Fgf21^-/-^* morulae. In the ensuing chimeras, 13del HCs are marked by EGFP expression, and express *Fgf21*. Mice with different degrees of *13del;Col10a1^Egfp^:13del;Fgf21^-/-^* chimerism, were analysed for HC survival. We found that *13del;Fgf21^-/-^* HCsadjacent to *13del;Col10a1^EGFP^*HCs expressing FGF21 still underwent apoptosis. In Figure S9D-F, we also show that the differentiation defect in 13del HCs was unchanged in HCs unable to express *Fgf21* at p10, indicating the observed cell fate change was FGF21-independent. These results are described in subsection “ATF4 and CHOP mediate chondrocyte survival by activating *Fgf21”*.

This inability of *13del;Col10a1^Egfp^* to rescue *13del;Fgf21^-/-^* HCssuggests that FGF21 works in a cell autonomous manner to protect the cell from apoptosis (new Figure 7C-E). We further show that *Fgf21* null mice have normal growth plates and HC viability (Figure 7—figure supplement 1). These results are described in subsection “ATF4 and CHOP mediate chondrocyte survival by activating *Fgf21”*

2) Redo the statistics using ANOVA rather than t- tests, which are not appropriate.

Thank you for the comment and advice. We have used Two-tailed Mann-Whitney *U*-test to test for statistical significance in all mouse phenotypic analyses. This is a nonparametric test of the null hypothesis that it is equally likely that a randomly selected value from one sample will be less than or greater than a randomly selected value from a second sample. Using this test, there is no change in outcome. We used Unpaired Two-tailed Student’s *t*-test in the luciferase assay.

3) Specify the selectivity of ISRIB and how it was solubilized; also address whether the authors used it in older mice at skeletal maturity.

The specific action of ISIRIB in targeting the translational control of p-eIF2α has been reported by different studies viz neurodegeneration (Dever et al., 1992; Harding et al., 2003), prostate cancer (Ron, 2002), memory enhancing (Wek, Jiang and Anthony, 2006; García, Meurs and Esteban, 2007), cognitive deficits after traumatic brain injury (Rzymski et al., 2010) and in rescuing of impaired sociability and anxiety-like behaviour (Ye et al., 2010). Biochemically, via a cell-base screen for inhibitors or PERK signaling, ISRIB (Integrated Stress Response Inhibitor) was identified as an effective inhibitor of ISR, that potently (IC_50_=5nM) reverses the effects of eIF2α phosphorylation (Wek, Jiang and Anthony, 2006).

In our paper we show ISRIB selectively functions as an inhibitor of ATF4 induction under ER-stress induced PERK signalling pathway (Figure 9). Consistent with the specific action on the PERK arm of the UPR, the ER stress induced degradation pathway mediated by IRE1, is not affected, as reflected by the unchanged expression level of XBP1^S^, as well as the persistent intracellular accumulation of 13del mutant protein.

In this study, the ISRIB was dissolved in DMSO to make a stock solution of 5mg/ml which was freshly diluted 10-fold with saline, for daily injection. We agree that the solubility of ISRIB has room for improvement, which may give a better therapeutic efficacy. In a recent prostate cancer study, we note that HPMT vehicle solution (0.5% w/v hydroxypropyl-methylcellulose dissolved in water plus 0.2% v/v Tween80, adjusted to pH 4) was used with favourable effect (Ron, 2002). This point has been discussed in the revised manuscript (Discussion section).

In our current study we have focused on using ISRIB to identify the upstream cause of the chondrocyte differentiation defect and also to test the utility of ISRIB in ameliorating or preventing the skeletal defect. We agree it would be interesting to know if treating mature MCDS mice with ISRIB could have utility in arresting or reversing the dwarfism but such study would take a long time and is outside the scope of our current study. It should be noted however that the 13del transgene reduces postnatally around 3 weeks and stops expressing in HCs at around 4-wk (Brostrom et al., 1996). We previously showed that the intensity of ER stress signalling correlates with levels of 13del expression (Brostrom et al., 1996). This would mean that in a scenario where the ER stress is relieved, such experiment on 13del mice may not be informative. Certainly, prevention of aberrant HC differentiation during the growth phase should be more effective therapeutically than once growth has ceased in maturity. This point has been discussed in the revised manuscript (Discussion section).

4) Clarify mouse backgrounds, sex, numbers and age.

In this study, we compared 13del vs. *13del;Ddit3^-/-^*, 13del vs. *13del;Sox9^c/c^, C10-Atf4* vs. *C10-Atf4;Sox9^c/c^*and *13del* vs. *13del; Fgf21^-/-^*. These comparisons were conducted between littermates only, so that any animal of desired genotype is only compared to its control littermates Furthermore, for each compound double or triple mutant mice, we used mice that had been inbred for >10 generation before any evaluation, to avoid genetic background issues. For clarification, we have described the genetic background and sampling methodology in the revised manuscript (subsection “Genetically modified mice and mutant analysis”). We did not observe any sex-associated phenotypic differences in MCDS, and the sex of test animals was randomized in this study. The number and age of test animals are now clarified in the corresponding figure legends.

Reviewer #1:[…] Of major importance there is no sex mentioned for the mice; the mouse numbers are small (n=5) and the mouse backgrounds are quite different. All this may contribute to some phenotypic differences; The Chop-/- mice are either 129 or CD1 mixed backgrounds, crossed with the 13del which are B6xCBA F1 hybrids; and then the crosses with FGF21 -/- that are on a B6 background. And there are no data on the *C10-Atf4* Tg. So much more explanation of the crosses is required and how the mixed background that result from these crosses could contribute to the growth plate phenotypes.

Please see our response to Essential Revision #4. The impact of mouse background has been minimized and the sex of test animals was randomized, given the fact that no sex-associated phenotypic changes were observed in MCDS.

FGF21 plays a critical role in the mutational phenotype in respect to survival; It would be useful to show the absence of a growth plate phenotype in the FGF21 null mice. Also, for Figure 6D qRT-PCR should be shown rather than the relative expression by micro array.

As described in our response to Essential Revision #1 above, we have included the analysis of *Fgf21^-/-^* growth plates in Figure 7—figure supplement 1B and C. We observed no significant changes in histology or frequency of cell death in the *Fgf21* null growth plates.

We show the microarray data as it was from analyses of the data that we first noted that *Fgf21* expression was turned on in 13del HCs, and it is not expressed in WT HCs. This finding has been validated both by in situ hybridization, Western blot and immunohistochemistry (Figure 6). Therefore, we have not performed qRT-PCR analyses especially because not all 13del-HCs expressed *Fgf21* and measurements of absolute RNA levels in a bulk RNA scenario may not truly reflect levels of expression per cell.

It’s not clear from the figure if in the 13del mice only Sox 9 is upregulated; particularly relevant would be changes in Gli 2,3 which have promoter sites for binding by ATF4; moreover, were the gene expression changes checked by qRT-PCR or only by micro array; that needs clarification in the legend.

We have examined the microarray data and show the expression trend of GLIs in 13del in Figure 3—figure supplement 1A. No significant upregulation of *Gli2* or Gli3 was observed in 13del-LHCs. In the revised manuscript, we have validated this observation using qRT-PCR (Figure 3—figure supplement 1B) and elaborated on this point in the revised main text (subsection “ATF4 reprograms chondrocyte hypertrophy by directly activating *Sox9”*) and figure legend.

How specific is ISRIB as an inhibitor of the Integrated stress response?Also, the rescue of 13del wit ISRIB was not complete and as noted there was still mutant collagen X present. Can the authors provide some insight into other potential mechanisms for the incomplete skeletal rescue if indeed the major pathway is through PERK?

As described in our response above, the specificity of ISIRIB, as an inhibitor of ISR has been previously reported by different studies.

As discussed in the manuscript, the incomplete rescue could be caused not only by the incomplete clearance of intracellular mutant collagen X, but also by limited bioavailability of ISRIB because of its poor solubility in biologically compatible solution as pointed out by reviewer 2. In our study, the expression level of ATF4 after treatment was lowered but was still detectable, the rescue therefore was incomplete. To improve the efficacy of ISRIB, the HPMT vehicle solution (as discussed in response to comment 3) could be tested in future studies. We thank the reviewer for the comment and have further elaborated on all these points in the revised Discussion section.

The body length phenotype is of interest in the *C10-Atf4* Tg; is it due to increases in vertebrae growth plate; intervetebral discs? Can the authors elaborate on that phenotype.

Thank you for the question. The drawfism of *C10-Atf4* is correlated with the abnormalities in the growth plates of both the appendicular and axial skeleton (subsection “ATF4 expression in hypertrophic chondrocytes reprogrammes differentiation”). To illustrate this point, we have included histological analysis of *C10-Atf4* vertebrae in Figure 2—figure supplement 1C and D.

Reviewer #2:[…] 1) Some of the experiments are presented as single examples (Figure 2A and E, Figure 3A and C, Figure 6A-D, Figure 8) albeit sometimes at multiple time points. Are repeats available and, if so, can the reproducibility of these effects be displayed in a quantitative form with statistical analysis? This is especially relevant to Figure 8B where the data quality is less good than elsewhere in this manuscript.

Thank you for the comment. All these experiments were biologically repeated (n=3 or 5) and the representative figures were shown. Also, we have repeated the experiment in Figure 8B, and better images were presented in new Figure 9B.

2) Could the authors comment on any solubility issues with ISRIB that may have impacted its bioavailability? This drug is somewhat notorious for its difficulty to dissolve in biologically compatible solutions.

As discussed previously, the poor solubility of ISRIB might impact its in vivo bioavailability, since the expression level of ATF4 after treatment was lowered but still detectable in ISRIB-treated HCs, the rescue therefore was incomplete. Thank you for the comment. We have further elaborated all these points in revised Discussion section.

3) The wording "ATF4 was also insignificantly lowered subsection “The PERK-p-eIF2α signalling pathway is downregulated explicitly by treatment of ISRIB” should probably be changed if the statistical test showed no difference. There is no need for Atf4 mRNA to be affected since the effects of ISRIB would be on ATF4 translation rather than transcription.

Thank you for the comment. We’ve rephrased the sentence (subsection “The PERK-p-eIF2α signalling pathway is downregulated explicitly by treatment of ISRIB”).

4) I am not convinced that Student's t-tests, which are used frequently herein, are always appropriately used. Statistical advice ought to be sought, and at the very least adjustments made for multiple comparisons. Analysis of variance with a subsequent post-hoc test would seem more appropriate to this reviewer. This is especially true for Figure 8 but is relevant in several other experiments where t-tests alone appear to have been employed.

Thank you for the comment and advice. We have used Two-tailed Mann-Whitney *U-*test to test for statistical significance in all mouse phenotypic analyses and Unpaired Two-tailed Student’s *t*-test in the luciferase assay. Also, we have repeated the Western blot experiment in old Figure 8B, and better images are presented in new Figure 9B.

Reviewer #3:[…] The authors should provide details of how the growth plate chondrocytes were fractionated into 3 or 4 subpopulations before their use in the transcriptome analyses. Because the data are used to reach mechanistic implications, it is important to show that each fraction was in fact composed by a distinct subpopulation. This is particularly important for 13del samples; given that the mutant growth plates contained abnormal distribution of Sox9-, col2- and Ihh-expressing cells, the cells may have been more difficult to separate into distinct subpopulations if they had intermediate cell phenotypes (volume, shape, etc.).

Thank you for the comment. The fractionation methodology is described in our recent paper (Tan et al., 2018), and is now elaborated on in the revised Materials and methods section as below: In brief, chondrocyte sub-populations in transverse sections of the proximal tibia of 10-day-old WT or 13del mouse were obtained by cryosectioning. 5-micron sections were prepared and pooled into fractions consisting of 10 sections per fraction to ensure separation of each cell type in the growth plate. Samples were dissolved in Trizol reagent (Invitrogen) for RNA extraction. To guide the subdivision of chondrocyte populations into zones, every 10th section was examined histologically and 10% of the RNA isolated from sections selected at regular intervals was used for the detection of known growth plate markers by RT-PCR analyses (Figure 1—figure supplement 1B).

The data on FGF21 are of particular interest. Fgf21 was found to be the most up-regulated gene in 13del hypertrophic chondrocytes, and Fgf21 deletion in 13del mice led to increased apoptosis in the hypertrophic zone. The authors suggest that the excess levels of FGF21 protected the 13del mutant chondrocytes from apoptosis. Do the authors have data on how such protection occurred and whether the fate of hypertrophic cells was altered in the compound mutant mice, including possible consequences on primary spongiosa? Based on data presented, it appears that FGF21 has no role in WT growth plates. Could the authors comment on this?

Please refer to our detailed response to Essential Revision #1.

The apparent beneficial effects of the ISR inhibitor on the 13del skeletal phenotype are promising, but appear to be modest. Also, only juvenile mice (up to p20) were examined. Did the authors try different drug regimens or administration routes to improve efficacy? Could beneficial effects be elicited in older mice up to skeletal maturity around 2-3 months of age? If treatment was stopped at P20 and mutant mice were followed over time, what happened to their skeletal phenotype?

Thank you for raising up these critical points for optimizing the preclinical study of ISRIB in MCDS treatment, which are also our current research goals. The mice at different stages will be studied and followed-up for different time periods. Please note our response in Essential Revision #3 above to improve the vivo bioavailability of ISRIB and testing ISRIB in older mice. Please note that 13del transgene expression is greatly diminished at around 4 weeks. However by that age the dwarfism is well established and the 13del mice remain dwarfed. This is consistent with a permanent impact of the ER stress during the developmental period and the phase of rapid postnatal growth mediated by the growth plate. Therefore after growth slows, the level of ER stress should be reduced and one would not expect any further change. This point has been discussed in the revised manuscript (Discussion section).

Reviewer #4: […] When Atf4 is overexpressed, Col10a1 expression is greatly decreased. How is the HC zone actually defined in this model? Why is *C10-Atf4* not itself downregulated?

*C10-Atf4* is a BAC transgene in which *Atf4* is inserted into *Col10a1* but is controlled by the regulatory elements of the *Col10a1* gene. We have shown that the BAC transgene recapitulates the HC specificity of expression. However, *Col10-Atf4* was not downregulated probably because the transgene may have integrated into a transcriptionally active site and could be in multiple copies, hence the expression of ATF4 is robust. However, in these transgenic mice the expression level of the endogenous *Col10a1* gene (diagnostic of the HC identity) is decreased in *C10-Atf4* mice because this is a reflection of the abnormal hypertrophic chondrocyte differentiation caused by ATF4 over-expression. However, since its expression was not totally abolished, we used this to help us to define the HZ.

If chondrocytes have impaired differentiation, does this mean that they retain features of proliferating or prehypertrophic chondrocytes? Does chondrocyte proliferation extend into this undifferentiated and expanded HC zone?

Thank you for the comment. We have performed BrdU labelling analysis on *C10-Atf4* mice (two hours before sacrifice) and observed that these *C10-Atf4* HCs were BrdU-negative (Figure 2—figure supplement 1F) (subsection “ATF4 reprograms chondrocyte hypertrophy by directly activating *Sox9”*). Given the fact that these cells express the pHC markers, such as *Sox9, Ihh* and *Ppr*, we conclude that these cells are prehypertrophic-like chondrocytes.

The expression of Sox9 could be quantified in Figure 5C. Is expression per cell changed or just number of cells?

The expression of *Sox9*, and its downstream target *Col2a1*, were measured via counting the staining-positive cells in growth plates. These results are shown in Figure 5E.

Figure 6D does not make sense. The text says 40% decrease in Fgf21 in 13del; CHOP-/- mice; however, the red line is above the green line in the figure?

Thank you for the comment. We have corrected this labelling mistake.

The potential role of FGF21 in cell survival is interesting; however, given the long range signaling abilities of FGF21, it should be shown whether FGF21 is acting locally on HCs or indirectly through another factor and non HC tissue. Is beta-klotho expression changed in HCs and is FGFR1 responsible for supporting cell survival in 13del HCs? Given that Fgf21 was highly increased, could it have effects other that mediating the ISR? Does FGF21 regulate growth hormone signaling through STAT5, IGF-1, or IGF-1 binding protein (see PMC2575072, PMC3406689).Although probably beyond the scope of this study, it may be interesting to see how the 13del mouse responds to FGF21 treatment at the growth plate. Is there a direct effect on ERK phosphorylation?

Please see our response in Essential Revision #1 above. Interestingly, HCs do not express beta-klotho and the expression level of STAT5, IGF-1 and IGF-1 binding protein was not changed in 13del HCs, examined by microarray (data not shown), suggesting that the pro-survival role of FGF21 in ER stressed HCs could be noncanonical. We have discussed these points in the revised manuscript (Discussion section).

Figure 8B Blot is of low quality. The actin bands are not consistent. This could explain the differences seen between UHC and LHC. Quantification of this expression data would help especially for BIP.

Thank you for the comment. We have repeated the experiment, and better images were presented in new Figure 9B.

Figure 9. It would be helpful to also indicate the suppressive effect of ISRIB on eIF2B and subsequently ATF4.

Thank you for the comment. We have added the mechanism of ISRIB action on eIF2B and subsequently on ATF4 in new Figure 10.

Figure 9—figure supplement 1D. Label indicates that TUNEL is Red, however the probe is Green. Change for consistency.

Thank you for the comment. We have changed the label colour to green.

[Editors' note: further revisions were requested prior to acceptance, as described below.]

The manuscript has been improved and the response to reviewers is complete except for the statistical analysis. This is the only remaining issue prior to acceptance and we ask that the authors more fully justify the use of the Mann-U Whitney test, particularly as it relates to multiple comparisons in Figure 8 vs the more standard and accepted one way or in some cases two way ANOVA. We recommend that you consult a statistician for your response. Once this is resubmitted, we can rapidly make a final decision.

Thank you for the comment and advice. We have consulted a biostatistics expert. He pointed out that ANOVA (or t-test) is very robust to non-normality as long as the group sizes are close to being equal. He would only advise the use of Mann-Whitney when there is very gross violation of normality (e.g. clear outlying observations) and when the group sizes are grossly unequal. In our data the groups being compared are of similar size and there are no clear outliers. Thus, we expected ANOVA to be robust. Indeed, when we performed ANOVA on our ISRIB treatment data, we found that all the results are very similar to our previous results from Mann-Whitney. As ANOVA is the more commonly used procedure and has the advantage that it can be used to compare 3 or more groups in a single test, the expert considers it reasonable to present the ANOVA results only. Thus, following both the reviewer’s suggestion and our expertise advice, we have presented the p-values obtained from ANOVA in Figure 8 and Figure 8—figure supplement 3.